# Multifeature fusion for claim scope-aware litigation risk prediction for patent drafts

Chitrakala Sakthivel* and Jinesh Jose*

Department of Computer Science and Engineering, College of Engineering Guindy Campus,
Anna University, Chennai, Tamil Nadu, India
* These authors contributed equally to this work.



## ABSTRACT

The 'claim scope', or the 'legal boundaries' defined by patent claims, has been considered crucial for determining a patent's value and its associated litigation risk. However, no direct claim semantics-based indicators currently exist to quantify patent claim scope, and existing scope measures are primarily indirect, which limits their ability to capture the semantic nuances of claim text. Additionally, the reliance on post-grant features restricts the applicability of existing litigation prediction models to patent drafts. These limitations complicate the patent drafting process, during which claims are formulated without feedback on scope and litigation risk. This often leads to suboptimal claim articulation, resulting in inadequate protection, increased legal vulnerabilities, or reduced patent grant probability. To address this gap, the hyponym tree score (HTS) is proposed as a novel indicator for quantifying claim scope by analysing hyponym counts, sentence structure, and dependency relations within patent claims. Building on this, early-stage litigation risk prediction has been achieved using a new deep learning model, the Multifeature BERT-Powered Fusion for Author-level Patent Litigation Risk Analysis (MAPRA). The MAPRA model restricts its input features to those available at early stages, such as indicators derived from claim text, inventor information, assignee details, and HTS, ensuring applicability to both draft-stage and granted patents. Despite excluding all post-grant or acquired data, MAPRA achieves a superior area under the receiver operating characteristic curve (AUC) of 0.878, outperforming the most comparable prior study, which reports an AUC of 0.822 using both early-stage and immediate post-grant features. By quantifying claim scope and enabling early-stage litigation risk prediction, this research offers a valuable screening tool for patent drafters, examiners, attorneys, and innovators. It supports informed decision-making during drafting and helps mitigate potential litigation risks. Furthermore, it lays a foundation for future research on claim scope modeling and the development of predictive tools for intellectual property litigation management.

## INTRODUCTION

The scope or coverage of a patent is defined by its claims, which establishes the boundaries of legal protection and serves as a critical determinant of the patent's enforceability, value, and commercial significance. Broader claim scope increases legal coverage and enhances

Corresponding author
Jinesh Jose, jj.india@gmail.com

the patent's market value but also raises the risk of conflicts with existing patents, thereby increasing the likelihood of litigation (*Merges & Nelson, 1994*; *Arinas, 2012*; *Marco, Sarnoff & Charles, 2019*). Conversely, narrower claim scope minimizes conflicts and improves the probability of a patent grant, but it may reduce the patent's legal coverage and economic potential (*Cotropia, 2005*; *Marco, Sarnoff & Charles, 2019*). Therefore, during the drafting of the patent claims, achieving an optimal balance in claim scope is essential to ensure robust legal protection, maximize patent value, and minimize litigation risks (*Tekic & Kukolj, 2013*).

Despite its importance, drafting patent claims remains complex and challenging, largely due to the absence of well-established, semantically rooted indicators for quantifying claim scope. Existing scope indicators often rely on bibliographic or numerical data and fail to incorporate the semantics of the claim text. This omission leaves patent drafters without clear guidance, leading to suboptimal claim articulation that may result in inadequate protection, heightened legal vulnerabilities, or unnecessary litigation risks. Addressing this challenge necessitates a robust, semantics-based metric that can quantify claim scope and aid drafters in achieving an optimal balance between legal coverage and litigation risk.

To fill this gap, the study introduces the hyponym tree score (HTS), a novel semantics-based indicator for quantifying the scope of patent claims. HTS leverages semantic relationships within the claim text, including hyponyms, sentence structures, and interdependencies between claims, to provide a meaningful and quantifiable measure of claim scope. By incorporating text semantics, HTS offers patent drafters actionable insights to optimize claim articulation, enhance legal protection, and mitigate litigation risks.

Patent litigation, which involves resolving disputes over patent infringement, validity, or enforcement, is critical in determining a patent's enforceability and commercial value. Litigation significantly influences market competition and potential revenue streams, underscoring its importance in the intellectual property landscape (*Helmers, 2018*). Predicting the likelihood of litigation is a key priority for stakeholders such as portfolio managers, insurers, patent valuators and patent drafters, as it enables strategic planning and effective risk management. Moreover, the articulation of claim scope is intricately linked to litigation risk, as broader claims are more likely to conflict with existing patents. In contrast, narrower claims may limit legal coverage (*Merges & Nelson, 1990*; *Marco, Sarnoff & Charles, 2019*).

Existing approaches to litigation prediction, however, face notable limitations. Prior studies have predominantly relied on post-grant event data and externally compiled features, such as the organisation for economic co-operation and development (OECD) patent quality indicators (PQI) (*Squicciarini, Dernis & Criscuolo, 2013*), available only for granted patents. These models are unsuitable for draft-stage patent documents, where such features are unavailable. Furthermore, many of these models neglect the semantic content of patent claims despite their critical importance in understanding the boundaries of patent protection and accurately predicting litigation risk.

This study proposes a novel multifeature fusion deep learning model for litigation prediction to overcome these limitations. Unlike existing models, this approach integrates

HTS with other features available at the drafting stage, making it applicable to both draft and granted patents. By relying exclusively on pre-grant features, the proposed model broadens the applicability of litigation prediction to include early-stage patent documents, empowering stakeholders to assess litigation risks at any stage of the patenting process.

This study significantly contributes to patent scope analysis and litigation prediction. First, it introduces the HTS, a semantics-based metric for quantifying claim scope, providing patent drafters with a valuable indicator for optimizing claim articulation. Second, it develops a self-sufficient multifeature fusion deep learning model for litigation prediction, designed to work with features available during the draft stage, thus addressing the limitations of existing litigation prediction models that rely on post-grant data. By bridging critical gaps in patent drafting and litigation prediction, this work represents a significant step forward in improving claim drafting, enhancing decision-making, improving strategic planning, and optimizing outcomes in the intellectual property domain. This research represents the first effort dedicated to predicting the litigation risk of the early-stage patent document.

## Overview of the article structure

The structure of this article is organized as follows: "Background" gives an overview of the context of this work. "Literature Review" reviews the relevant literature and identifies gaps this study aims to address. "Methodology" details the methodology, including data collection ("Dataset"), the development of the hyponym-based indicator ("Claim Scope Indicator Development"), and the development of the new deep learning model for litigation prediction ("Litigation Prediction Model Development"). "Results" presents the results of this study, including the performance of the new litigation prediction model and the relevance of the hyponym-based claim scope indicator. "Discussion" discusses the findings, implications, and potential limitations. Finally, "Conclusion" concludes the article with a summary of key insights and suggestions for future research.

## BACKGROUND

The research originates from an ongoing investigation into developing robust models for patent valuation. A notable trend was observed during the investigation: high-value patents are more likely to face legal events and litigation proceedings (*Tekic & Kukolj, 2013*). This finding raised interest in predicting patent litigation, particularly for early-stage patent documents, by leveraging machine learning techniques to forecast the likelihood of legal disputes. Even though claim text semantics play a pivotal role in defining the scope or coverage of a patent, the absence of a measure to quantify the claim scope impedes the drafters from optimally regulating the claim scope during the claim drafting. Additionally, understanding the litigation risk of a patent during the drafting stage allows professionals to regulate claim scope effectively by choosing appropriate wording. Developing a litigation prediction model that relies solely on pre-grant patent features can enable litigation risk prediction for both granted and early-stage patent documents.

## Literature review

This subsection presents a comprehensive review of the relevant literature, organized into two main areas: (1) Indicators of patent scope and (2) patent litigation prediction models. Each group is critically analysed to identify existing limitations and to highlight how this study fills the identified gaps.

### Patent scope indicators

Quantifying the scope of a patent is a longstanding challenge in intellectual property research. Numerous indicators have been proposed to estimate the breadth of legal protection and technological applicability offered by patents. These can be categorized into the following groups:

**Citation-based indicators:** Citation analysis has been extensively utilized in patent research, primarily through backward and forward citation metrics. The number of forward citations, originally proposed by *Trajtenberg (1990)*, is widely used to assess a patent's technological impact, with a higher number generally interpreted as reflecting broader scope. The number of backward citations indicates the extent of prior art reviewed, suggesting a wide technological foundation (*Packalen & Bhattacharya, 2012*). In addition, non-patent literature (NPL) citations indicate a broader research base supporting the invention, as noted by *Narin, Hamilton & Olivastro (1997)*. However, forward citations are not available for early-stage or draft patents, limiting their practical utility during the drafting phase.

**Patent classification-based indicators:** Classification-based indicators assess technological breadth based on the number of categories assigned to a patent. Studies by *Lerner (1994)* and *Harhoff, Scherer & Vopel (2003)* have demonstrated that patents with a greater number of subclasses tend to span a wider array of technological fields, reflecting broader applications and scope.

**Claim-based indicators:** Claim-based indicators are among the most direct measures of patent scope and can be further divided into two subgroups: indicators based on claim quantity and those based on claim structure.

Claim quantity-related indicators focus on the number and types of claims included in the patent. The number of claims is widely recognized as a measure of scope, with a greater quantity generally suggesting broader protection (*Lanjouw & Schankerman, 1997*, *2001*, *2004*). Similarly, the number of independent claims is interpreted as reflecting wider coverage, since each independent claim typically represents a distinct technological aspect (*Marco, Sarnoff & Charles, 2019*; *Graham & Mowery, 2003*). In contrast, dependent claims, although providing specificity and detail, do not significantly contribute to a broader scope (*Graham & Mowery, 2003*).

Claim structure-related indicators evaluate the linguistic, syntactic, and logical organization of individual claims. Commonly used metrics include words per claim (*Lerner, 1994*; *Osenga, 2011*; *Harhoff, 2016*), independent claim length (*Malackowski & Barney, 2008*; *Marco, Sarnoff & Charles, 2019*), and first claim length (*Harhoff, 2016*;

*Wittfoth, 2019*). These studies have suggested that shorter claims are generally broader in scope due to fewer embedded limitations. *Okada, Naito & Nagaoka (2016)* introduced character count as an alternative metric, particularly useful in languages without word spacing, arguing that longer character sequences correlate with narrower, more detailed claims. Additionally, claim dependency structure, as explored by *Wittfoth (2019)*, plays a role in defining the hierarchical and interpretive relationship between independent and dependent claims, impacting how broadly a claim set may be interpreted.

**Semantics-based indicators:** In response to limitations of numeric and bibliographic features, recent studies have introduced semantics-driven approaches. *Tanaka, Nakashio & Kajikawa (2018)* proposed the use of semantic range of words to measure vocabulary diversity, enabling scope visualization through semantic hierarchies. *Ragot (2023)* introduced a novel textual metric called self-information, which quantifies the informativeness of individual claims. Their findings suggest that higher self-information scores correlate with broader conceptual scope.

The number of inventors has also been interpreted as an indirect scope metric. *Chan, Mihm & Sosa (2021)*, highlighted that a higher number of inventors reflects greater collaboration and the non-decomposability of the invention. The scope tends to decrease with the number of inventors.

Table 1 summarizes existing scope indicators, outlining their theoretical bases and known limitations. While these metrics span a range of approaches, they predominantly rely on bibliographic data, numeric heuristics, or surface-level linguistic cues. Notably absent are robust, semantically informed indicators capable of evaluating the breadth of a patent claim based on its underlying meaning and hierarchical structure. Currently, no widely adopted method allows authors to quantify whether a claim is semantically broad or narrow. This gap hinders precise calibration of claim scope and increases the risk of either under-protecting the invention or inviting legal challenges due to overly broad claims.

### *Patent litigation prediction models*

The prediction of patent litigation has evolved substantially, transitioning from traditional statistical models to sophisticated machine learning (ML) and deep learning (DL) frameworks. The existing literature can be organized into the following categories:

**Classical machine learning approaches:** Early work in this area focused on regression-based and tree-based models. *Chien (2011)* employed logistic regression (LR) to analyze how specific intrinsic and acquired patent traits influence the likelihood of litigation. *Juranek & Otneim (2021)* used the XGBoost algorithm with features drawn from united states patent and trademark office (USPTO) datasets, OECD patent quality indicators (PQI), and USPTO patent litigation docket reports, achieving high predictive performance (AUC up to 0.818). They identified that indicators related to patent value, internationality, and patent owner characteristics hold higher predictive power. However, their model's reliance on post-grant information limits its applicability during the drafting phase. Similarly, *Follesø & Kaminski (2020)* utilized random forest (RF) classifiers trained on PQI-derived features to assess litigation risk.

**Table 1 Summary of the patent scope indicators.**

| Scope indicator | Literature | Remarks |
|---|---|---|
| Number of forward citations | *Trajtenberg (1990)* | More forward citations reflect greater impact and scope. |
| Number of claims | *Lanjouw & Schankerman (1997, 2001, 2004)* | More claims suggest broader scope. |
| Number of NPL citations | *Narin, Hamilton & Olivastro (1997)* | More citations to non-patent literature imply a broader research base. |
| Words per claim | *Lerner (1994), Osenga (2011), Harhoff (2016)* | Shorter claims indicate broader coverage. |
| Number of sub classes | *Lerner (1994), Harhoff, Scherer & Vopel (2003)* | More subclasses indicate technological diversity. |
| Number of independent claims | *Marco, Sarnoff & Charles, 2019, Graham & Mowery (2003)* | More independent claims mean broader scope. |
| Number of dependent claims | *Graham & Mowery (2003)* | More dependent claims provide detailed extensions of the main invention. |
| Independent claim length | *Malackowski & Barney (2008), Marco, Sarnoff & Charles (2019)* | Shorter independent claims indicate broader coverage. |
| Number of backward citations | *Packalen & Bhattacharya (2012)* | More backward citations reflect wider prior art. |
| First claim length | *Harhoff (2016), Wittfoth (2019)* | Shorter first claims are broader. |
| Claim's character count | *Okada, Naito & Nagaoka (2016)* | More characters suggest a narrower scope. |
| Semantic range of words | *Tanaka, Nakashio & Kajikawa (2018)* | Reciprocal of the number of hierarchies is considered |
| Based on dependencies of independent and dependent claims | *Wittfoth (2019)* | Dependency structure affects the scope of the claims. |
| Number of inventors | *Chan, Mihm & Sosa (2021)* | More inventors indicate higher collaboration and non-decomposable invention. |
| Self-information | *Ragot (2023)* | Quantifies unique information each claim provides. |

**Semantic and similarity-based approaches:** Several researchers have explored textual content to infer litigation potential. *Park, Yoon & Kim (2012)* applied semantic similarity analysis based on Subject-Action-Object (SAO) patterns and clustering to identify potential infringement scenarios. *Lee, Song & Park (2013)* evaluated claim text similarity using keyword vector models and analysed inter-claim dependencies. While these methods incorporate both linguistic and structural elements, they often face limitations in scalability and generalizability, particularly across large or heterogeneous patent datasets. Although effective in identifying potential overlaps between pairs of patents, extending such analysis to all patent pairs for litigation prediction poses significant computational challenges.

**Unsupervised and ensemble techniques:** Several studies have integrated unsupervised learning and ensemble methods to enhance prediction accuracy. *Wongchaisuwat, Klabjan & McGinnis (2017)* combined K-means clustering with ensemble classification models to estimate the likelihood and timing of litigation jointly. *Kim et al. (2022)* applied principal component analysis (PCA) for dimensionality reduction and used Autoencoders in combination with K-nearest neighbors (K-NN) for classification, improving predictive performance by emphasizing the most informative features. *Chen & Lai (2023)* implemented an ensemble machine learning classifier leveraging USPTO examination and assignment data, achieving 79% accuracy and demonstrating the viability of ensemble methods for litigation risk assessment.

**Deep learning models:** Recent advances in deep learning have enabled the modeling of complex, multi-dimensional relationships present in patent litigation data. *Liu et al. (2018)* proposed a convolutional tensor factorization framework to identify high-risk patents based on textual and collaboration features. *Wu et al. (2024)* introduced the multi-aspect neural tensor factorization (MANTF) model to predict plaintiffs, defendants, and target patents jointly. Convolutional neural networks (CNNs) have also been utilized for one-to-many infringement detection (*Liu & Pei, 2023*), while *Kim et al. (2021)* employed random survival forests to model litigation risk over time.

The most recent and closely related work to the objectives of this study is by *Juranek & Otneim (2024)*, who refined their XGBoost model to handle newly granted patents by minimizing reliance on post-grant features that are not available at the time of grant. In their study, the XGBoost algorithm was used for litigation prediction and achieved an AUC score of up to 0.822. However, this approach remains inapplicable to draft-stage documents due to its dependence on post-grant data.

Table 2 summarizes the prominent litigation prediction models and related studies, outlining their methodological foundations and known limitations. While these approaches span a range of machine learning and deep learning techniques, the majority rely on post-grant features such as forward citations, patent family size, assignment records, and other patent quality indicators. Models that assess litigation risk using only information available at the drafting stage are notably absent from the existing literature. In particular, semantic features embedded within patent claims, despite being central to legal interpretation and enforceability, remain largely underutilized in current predictive frameworks. Although some studies have applied semantic similarity analysis to identify potential overlaps or infringement between individual patent pairs, scaling such analyses across large patent datasets introduces significant computational challenges. Furthermore, no existing model provides a structured framework for predicting litigation risk at the draft stage using claim-level semantic features. This gap restricts the ability to conduct early-stage risk assessment and reduces the practical value of these models for inventors, legal professionals, and innovation strategists. The literature survey indicates that the proposed work is a pioneering effort for litigation prediction in patent drafts, and no comparable work for a one-to-one comparison is available.

## Research gaps

Current scope indicators for patents can be broadly categorized into pre-grant and post-grant indicators based on their availability. For instance, indicators like the 'number of claims' and 'backward citations' are accessible during the pre-grant stage. In contrast, indicators such as 'forward citations' and 'grant lag' become available only after a patent is granted. Relying on indicators available at the pre-grant stage is crucial for assessing the scope of early-stage patent documents. As depicted in Table 1, established scope indicators do not focus on the semantics of the claim text when determining the patent scope. The lack of well-established claim scope indicators rooted in claim text semantics complicates the drafting process, frequently leading to suboptimal articulation of claim scope. This

**Table 2 Summary of the litigation prediction works.**

| Authors | Recommended method | Remarks |
|---|---|---|
| Chien (2011) | Logistic regression | Analyses the impact of intrinsic and acquired traits of patents in litigation |
| Park, Yoon & Kim (2012) | SAO-based semantic similarity measurement and clustering | SOA-based semantic technological similarity are computed between each patent, and clustering is applied to identify the clusters of patents with possible infringements. |
| Lee, Song & Park (2013) | Statistical methods (t-statistics, critical mean value) and hit ratios. | Similarity between all the patents are calculated based on keyword vectors and claim interdependence |
| Wongchaisuwat, Klabjan & McGinnis (2017) | K-means clustering and ensemble classification. | Predicts the litigation likelihood and the expected time to litigation |
| Liu et al. (2018) | Convolutional Tensor Factorization | Helps to identify the risky patents using their content and collaborative information |
| Follesø & Kaminski (2020) | Random forest | Litigation Prediction using OECD PQI features |
| Kim et al. (2021) | Clustering and random survival forest | Predicts patent litigation risk over time and considers the censored data |
| Juranek & Otneim (2021) | XGBoost | Features from different data sets provided by the USPTO, Patent Litigation Docket Reports Data & OECD PQI are used. 0.818 AUC reported with XGBoost. |
| Kim et al. (2022) | K-NN and autoencoder | PCA based feature extraction on quantitative features |
| Wu et al. (2024) | Multi-aspect neural tensor factorization | Can predict potential plaintiffs, defendants and patents |
| Chen & Lai (2023) | Ensemble machine learning classifier | Uses examination and assignment data and reported 79% accuracy |
| Liu & Pei (2023) | CNN | One to many infringement detection |
| Juranek & Otneim (2024) | XGBoost | Restricted to the features available at the time of grant. Indicators related to value, inter-nationality and patent-owners have higher predictive power. 0.822 AUC Reported with XGBoost. |

deficiency may lead to future financial losses due to insufficient protection or excessive legal costs associated with overly broad claims.

The current research on patent litigation prediction predominantly relies on externally compiled or post-grant features, such as international patent classification (IPC) details, forward citations, *etc.*, which are only available for granted patents. Such feature requirements make them unsuitable for performing the litigation prediction on draft stage documents for which such features are unavailable. Another notable observation is that current works predominantly neglect claim semantics, which define the legal boundaries. To expand the applicability of litigation prediction models to a broader range of patent documents, including those in the pre-grant stage, it is imperative to develop methods that use only the features available at the early stage.

## Research objectives

This study seeks to address the aforementioned gaps and advance the field of patent litigation prediction through the following objectives:

1. To develop the HTS, a novel metric to quantify the scope of patent claims by analyzing semantic relationships in claim text, leveraging hyponyms, sentence structures, and interdependencies among claims.

2. To design a multifeature fusion deep learning litigation prediction model that relies on claim text semantics and uses only early-stage features, ensuring applicability to both granted and draft-stage patent documents.

### Research questions

This study aims to address the following research questions:

RQ1 How can a semantics-based indicator be developed to quantify the scope of patent claim text?

RQ2 What is the impact of incorporating the new claim scope indicator on patent litigation prediction tasks?

RQ3 How can a high-performance litigation prediction model be developed to predict the litigation risk of draft-stage patent documents?

## METHODOLOGY

The development of a new indicator to quantify the patent claim scope and its evaluation using a litigation prediction task is presented in the first part of this work. The HTS is the proposed indicator. A litigation prediction model for draft-stage patent documents is developed in the second part. The proposed litigation prediction model is named Multifeature BERT-Powered Fusion for Author-level Patent Litigation Risk Analysis (MAPRA).

### Dataset

This study is based on four primary datasets, each contributing essential information for patent scope analysis and litigation prediction. The USPTO PatentsView dataset (*U.S. Patent and Trademark Office, 2024a*; *Toole, Jones & Madhavan, 2021*) serves as the primary source of patent data, offering information on classification codes, inventors, assignees, and claim text. The 2024 update of this dataset is utilized in the present work. Complementing this, the OECD PQI database, January 2024 version (*Organisation for Economic Co-operation and Development (OECD), 2024*; *Squicciarini, Dernis & Criscuolo, 2013*), provides quantitative indicators capturing various dimensions of patent quality, such as technological relevance and potential economic value. Although only pre-grant features are incorporated into the prediction models, select PQI indicators are employed to evaluate the HTS.

Litigation data are obtained from the USPTO Patent Litigation Dataset (*U.S. Patent and Trademark Office, 2024b*; *Toole, Miller & Sichelman, 2024*), which records U.S. district court cases involving patent disputes filed between 1963 and 2020. This dataset includes 56,488 unique litigated patents. After applying a series of preprocessing operations, including merging and filtering, the final set comprises 40,897 unique litigated patents, each linked to its claim text, IPC classifications, and other relevant features. Patents not

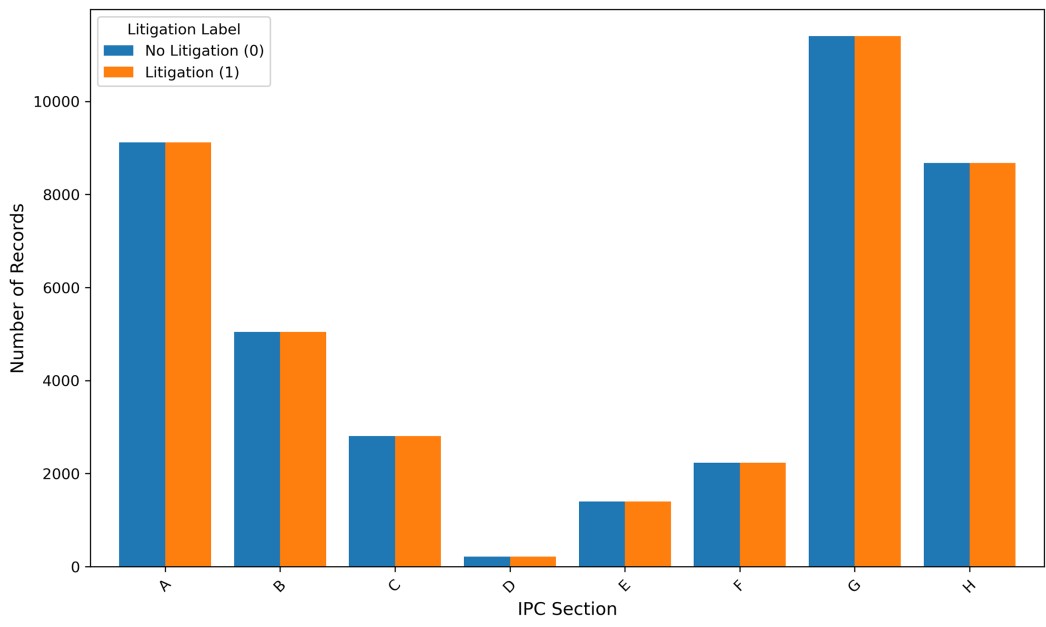

**Figure 1 Number of IPC sections in the sampled dataset.**

listed in the litigation dataset are treated as non-litigated as of 2020. However, to mitigate potential mislabeling due to delayed litigation, the sampling of non-litigated patents is restricted to those filed on or before 2010. This criterion ensures that most patents would have been granted by 2015, allowing for at least five years of post-grant observation within the litigation data collection window. Following established methodologies in the literature (*Juranek & Otneim, 2024*; *Liu, Li & Liu, 2024*), a total of 40,897 non-litigated patents are sampled to serve as the negative class. Patent litigation is a relatively rare event and affects fewer than 2% of all granted patents (*Chien, 2011*; *Wongchaisuwat, Klabjan & McGinnis, 2017*; *Juranek & Otneim, 2021*). Including all non-litigated patents would reflect real-world distributions but would also introduce substantial computational burdens, particularly for transformer-based models such as bidirectional encoder representations from transformers (BERT). To address this challenge, a 1:1 matched sampling strategy is employed.

As shown in Fig. 1, each litigated patent is paired with a non-litigated patent, resulting in a balanced dataset for training and evaluation. This approach aligns with the methodology adopted by *Park, Bhardwaj & Hsu (2023)*, who implemented matched sampling based on filing year and cooperative patent classification (CPC) subclass code in the context of robustly optimized BERT pretraining approach (RoBERTa) based litigation prediction (*Park, Bhardwaj & Hsu, 2023*). In the present work, non-litigated patents filed on or before 2010 are sampled to mirror the distribution of IPC sections found in the litigated patent set. The final dataset consists of 81,794 records, with an equal number of litigated and non-litigated patents. A detailed description of all variables, their sources, and their intended roles in the analysis is provided in Table 3.

**Table 3  Details of the features used in this study.**

| Feature | Data source | Description |
|---|---|---|
| bwd_cits | OECD PQI | Number of backward citations |
| npl_cits | OECD PQI | Number of non-patent literature backward citations |
| claims_x | OECD PQI | Number of claims |
| filing | OECD PQI | Year of filing |
| dependent_claims | PatentsView | Number of dependent claims, calculated from claim text |
| independent_claims | PatentsView | Number of independent claims, calculated from claim text |
| claim_text | PatentsView | Text containing all the patent claims |
| assignee_pcount | PatentsView | Number of patents owned by the assignee, calculated from Assignee data |
| num_inventors | PatentsView | Number of inventors |
| avg_claim_length | PatentsView | Average claim length, calculated value |
| fc_word_count | PatentsView | Number of words in the first claim, calculated from claim text |
| hts_spacy | Generated | Generated feature, not used in the final modal |
| hts_spacy_wtd | Generated | Generated feature, used in the final modal |
| hts_stanza | Generated | Generated feature, not used in the final modal |
| hts_stanza_wtd | Generated | Generated feature, not used in the final modal |
| hts_avg | Generated | Generated feature, not used in the final modal |
| hts_avg_wtd | Generated | Generated feature, not used in the final modal |
| litigation_label | Litigation Docket | Binary litigation status calculated using USPTO Litigation Docket Data |

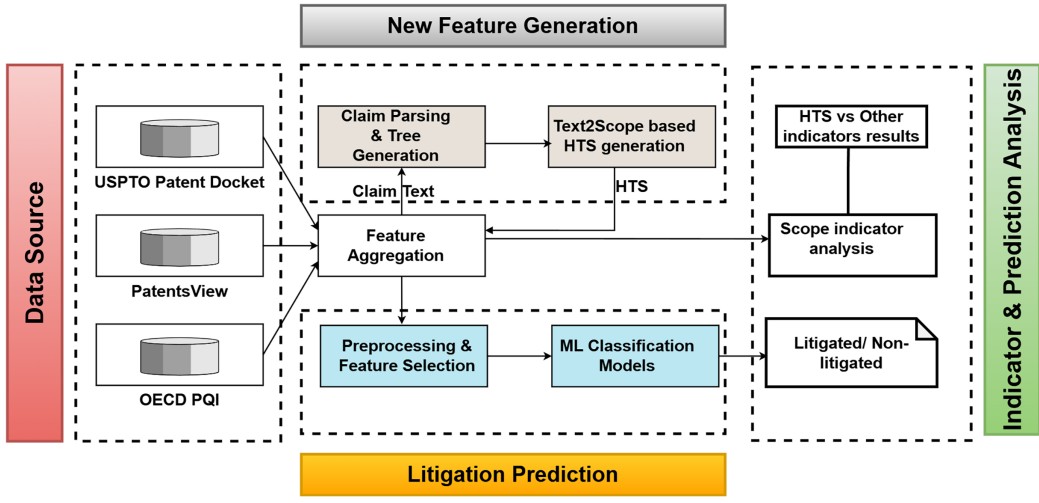

**Figure 2  Overview of the HTS feature generation and evaluation.**

## Part 1: claim scope indicator development

This part of the study aims to derive a new indicator to quantify the claim scope based on the claim text semantics. Figure 2 represents the high-level view of the work. Before determining an appropriate methodology for claim scope quantification, understanding the nature of the patent claim text is essential.

The claim text is a semi-structured text *corpus* with numbered claims, where each claim may explicitly reference other claim numbers to represent interconnections. Patent claims are broadly categorized into two types based on interdependency: independent and dependent claims. Independent claims are self-contained and provide a broad, comprehensive description of the invention, outlining its essential features without relying on other claims. These claims establish the widest boundaries of the patent's protection. Conversely, dependent claims refer back to independent claims and add specific features or details, resulting in a narrower scope of protection. Dependent claims serve as fallback positions if the independent claim is deemed invalid, ensuring that specific embodiments or variations of the invention remain safeguarded. A typical claim consists of three main components: the preamble, the transitional phrase, and the body. The preamble establishes the context of the claim by identifying the invention's category, such as a device, method, composition, or apparatus. The preamble aligns with the title of the invention and may include its objective or purpose. The transitional phrase links the preamble to the body, defining the claim's scope. Transitional phrases are categorized into open-ended, such as "comprising", which allows additional elements not explicitly mentioned in the claim, thereby broadening its scope, and closed-ended, such as "consisting of", which limits the claim strictly to the listed elements. The body of the claim is the most critical part, detailing the elements and limitations of the invention and describing their meaningful interconnections. The body provides an in-depth explanation of how the components interact to realize the invention, ensuring clarity and precision in defining the scope of protection.

### Hyponym tree score calculation

In natural language processing (NLP), hyponyms and hypernyms represent hierarchical relationships between words, which are crucial for understanding semantics and building structured knowledge. A hypernym refers to a broader, more general term, while a hyponym refers to a narrower, more specific term that falls under the hypernym. For example, in a taxonomy, 'vehicle' represents a hypernym of 'car', and 'car' serves as a hyponym of 'vehicle'. Similarly, the hypernym 'fruit' encompasses hyponyms such as 'berry', 'banana', and 'mango'. These relationships are often modelled in NLP using resources like WordNet (*Fellbaum, 1998*), where hypernym-hyponym hierarchies are explicitly defined. Understanding such relationships enables NLP systems to infer broader or narrower meanings, which is essential for analyzing the scope of patent claim texts. Words with more hyponyms in a patent claim indicate the potential to create multiple restrictive versions of claims, which can lead to overlaps in scope representing potential infringement cases and litigation risks. Consequently, studying hyponyms within patent claim text is pivotal in devising a new scope indicator for patents.

Patent claims often employ varying levels of specificity, where claims with broader scope support a larger interchangeability of terms to protect a more extensive set of derived ideas (*Cohen & Lemley, 2001*). However, when a claim employs overly generic language, the claim scope increases drastically, potentially clashing with more specific claims in other patents, leading to increased litigation risks and legal uncertainties. Conversely, highly

specific claims may reduce infringement risks but face challenges in enforcing their rights against variations and derivative innovations. Analyzing hypernym and hyponym characteristics within patent claim texts (*Andersson et al., 2014*) can potentially play a crucial role in claim scope quantification. In this context, a new HTS indicator is developed to represent the patent scope. The HTS indicator is derived by considering the hyponym count in claim sentences and their structural composition. When words in a patent claim text have more hyponyms, the possibility of interchangeability increases, broadening the claim scope. The new scope indicator will be validated by assessing its effectiveness in predicting patent litigation likelihood.

Mathematically, let the patent claim text be represented as a hyponym dependency tree, $T = (V, E)$, where $V = \{v_1, v_2, \ldots, v_n\}$ is the set of nodes corresponding to the words in the claim, and $E$ is the set of directed edges that denote the syntactic or semantic dependency relations between these words. Each node $v_i \in V$ is associated with a degree $\deg(v_i)$, representing the number of hyponyms (*i.e.*, more specific terms) that can replace the corresponding word. The degree $\deg(v_i)$ reflects the flexibility of the word within the claim text, where higher values represent greater possibilities for creating restrictive variations of the claim.

Given the tree structure, a cumulative score $C$ for the entire set of claims as follows:

$$C(T) = \prod_{v_i \in V} (\deg(v_i) + 1),$$

where $\deg(v_i)$ is the degree of node $v_i$, representing the number of hyponyms for the word corresponding to node $v_i$ and the term $(\deg(v_i) + 1)$ accounts for the word itself (original term) and its associated hyponyms.

This cumulative score reflects the maximum number of specific or restrictive versions of the claims that could be generated from the given claim text. Each restricted version is a modified claim with a smaller scope, offering different legal interpretations and enforcement potentials. The cumulative score $C(T)$ indicates the scope of the original patent claims. A larger claim scope increases the likelihood of overlapping with other patents, a primary cause of litigation. Patents with higher cumulative scores are more prone to infringement due to the more significant number of possible interpretations and restrictive variations that could overlap with existing claims. Thus, $C(T)$ can quantify the scope or coverage of the patent claim and provide a theoretical foundation for predicting patent litigation risk based on hyponym analysis. Multiplicative $C(T)$, which calculates the number of sub-trees possible from the original tree, has a problem with the lengthy claims producing very large values, and the effect of smaller claims goes unnoticed, hence discarded.

Three tasks were carried out to calculate the HTS of claim text: claim dependency tree generation, dependency tree generation for each sentence in the claim, and hyponym extraction of the words in each sentence. Algorithm 1, the Text2Scope, was developed to compute the HTS value from the patent claim text. Claims are represented as a graph with individual claims as the nodes and the dependency among them as the edges. A claims text *corpus* is processed using Algorithm 1 (Text2Scope), and a tree structure of the claims is

---

**Algorithm 1  Text2Scope: calculate the scope score for an entire patent claim tree.**

1: **Input:** Text of multiple claims
2: **Output:**
3: Cumulative scores for the entire claim tree:
4: *hts_spacy*, *hts_spacy_wtd*
5: Parse the text to extract individual claims, each with a claim number and text.
6: Initialize a directed graph *G* where:
7: Nodes represent claims, and edges represent references between claims.
8: Initialize variables:
9: $hts\_spacy \leftarrow 0$, $hts\_spacy\_wtd \leftarrow 0$, $connected\_components \leftarrow 0$.
10: **for** each claim **do**
11:      Identify references to other claims.
12:      Add the claim as a node in *G*.
13:      Add edges from the claim to referenced claims.
14: **end for**
15: Find connected components in *G*.
16: **for** each connected component *C* in *G* **do**
17:      Initialize $component\_hts\_spacy \leftarrow 0$, $component\_hts\_spacy\_wtd \leftarrow 0$,
         $group\_size \leftarrow 0$.
18:      **for** each claim in *C* **do**
19:          Apply `Claim2Scope` on the claim text to compute individual scores:
20:          Obtain *claim_hts_spacy* and *claim_hts_spacy_wtd*.
21:          Update $component\_hts\_spacy \leftarrow component\_hts\_spacy + claim\_hts\_spacy$.
22:          Update $component\_hts\_spacy\_wtd \leftarrow component\_hts\_spacy\_wtd +$
             $claim\_hts\_spacy\_wtd$.
23:          Increment $group\_size \leftarrow group\_size + 1$.
24:      **end for**
25:      Update $hts\_spacy \leftarrow hts\_spacy + \frac{component\_hts\_spacy}{\max(1, group\_size)}$.
26:      Update $hts\_spacy\_wtd \leftarrow hts\_spacy\_wtd + \frac{component\_hts\_spacy\_wtd}{\max(1, group\_size)}$.
27: **end for**
28: **return** *hts_spacy*, *hts_spacy_wtd*.

---

**Algorithm 2  Claim2Scope: calculate the scope score for a claim.**

1: **Input:** Claim text *p*
2: **Output:**
3: Claim score components:
4: *hts_spacy*, *hts_spacy_wtd*, *avg_tree_height*,
5: *total_h_count*, *total_h_sum*, *total_node_count*, *number_of_sentences*
6: Split *p* into individual sentences.
7: Initialize variables to accumulate scores and counts across sentences:
8: $hts\_spacy \leftarrow 0$, $hts\_spacy\_wtd \leftarrow 0$, $total\_h\_count \leftarrow 0$,

---

**Algorithm 2** (continued)

9:  $total\_h\_sum \leftarrow 0$, $total\_node\_count \leftarrow 0$,

10:  $total\_tree\_height \leftarrow 0$, $number\_of\_sentences \leftarrow 0$.

11: **for** each sentence $s$ in $p$ **do**

12:     Apply `Sentence2Scope` on $s$ to obtain:

13:     $hts\_spacy$, $hts\_spacy\_wtd$, $tree\_height$, $c\_node$, $h\_count$, $h\_sum$, $wtd\_h\_sum$.

14:     Update $hts\_spacy \leftarrow hts\_spacy + hts\_spacy$.

15:     Update $hts\_spacy\_wtd \leftarrow hts\_spacy\_wtd + hts\_spacy\_wtd$.

16:     Update $total\_h\_count \leftarrow total\_h\_count + h\_count$.

17:     Update $total\_h\_sum \leftarrow total\_h\_sum + h\_sum$.

18:     Update $total\_node\_count \leftarrow total\_node\_count + c\_node$.

19:     Update $total\_tree\_height \leftarrow total\_tree\_height + tree\_height$.

20:     Increment $number\_of\_sentences \leftarrow number\_of\_sentences + 1$.

21: **end for**

22: Calculate $avg\_tree\_height = \frac{total\_tree\_height}{\max(1, number\_of\_sentences)}$.

23: **return** $hts\_spacy$, $hts\_spacy\_wtd$, $avg\_tree\_height$,

24: **return** $total\_h\_count$, $total\_h\_sum$, $total\_node\_count$, $number\_of\_sentences$.

---

**Algorithm 3**   **Sentence2Scope: calculate the hyponym tree score for a sentence.**

1: **Input:** Sentence $s$

2: **Output:** Hyponym Tree Score $hts\_spacy$, Weighted Hyponym Tree Score $hts\_spacy\_wtd$, Tree Height $tree\_height$, Node Count $c\_node$, Hyponym Count $h\_count$, Hyponym Sum $h\_sum$, Weighted Hyponym Sum $wtd\_h\_sum$

3: Initialize directed graph $G$, root node `root_word` as `None`, and other variables.

4: Process the sentence $s$ to extract tokens using spaCy.

5: **for** each word $w$ in $s$ **do**

6:   **if** $w$ is not a stop word **then**

7:     Compute hyponym count $hyponyms\_count$ for $w$.

8:     Update $h\_count$ and $h\_sum$.

9:     Add $w$ as a node in $G$ with attributes (label, hyponyms_count).

10:   **end if**

11:   Add dependency relationships between tokens in $G$.

12:   **if** $w$ is the root of the dependency parse tree **then**

13:     Set `root_word` $\leftarrow w$.

14:   **end if**

15: **end for**

16: **if** $root\_word$ is `None` **then**

17:   **return** default values.

18: **end if**

19: Assign levels and weights to nodes in $G$ using a BFS traversal starting from `root_word`.

(Continued)

| Algorithm 3 (continued) |
|---|
| 20: Compute *wtd_h_sum* as the weighted sum of hyponym counts based on node levels. |
| 21: Compute *tree_height* as the maximum depth of G. |
| 22: Compute *c_node* as the total number of nodes in G. |
| 23: Compute *normalizer* = (*c_node* × *tree_height*). |
| 24: Compute $hts\_spacy = \frac{h\_sum}{normalizer}$. |
| 25: Compute $hts\_spacy\_wtd = \frac{wtd\_h\_sum}{normalizer}$. |
| 26: **return** *hts_spacy*, *hts_spacy_wtd*, *tree_height*, *c_node*, *h_count*, *h_sum*, *wtd_h_sum*. |

generated initially. In the tree, each node contains the text corresponding to a numbered claim. Algorithm 2 (Claim2Scope) is invoked from Text2Scope to calculate the score of a given patent claim. Claim2Scope invokes Algorithm 3 (Sentence2Scope) to calculate the score of each sentence of the given claim. Sentence2Scope algorithm involves dependency tree generation to extract the sentence structure and node weight assignment using the hyponym counts of each node(word). Then it computes the cumulative score calculation for that sentence. These algorithms return hyponym tree scores and weighted hyponym tree scores. Equation (1) represents the sentence level non-weighted score calculation. Equation (2) is used for weighted score calculation.

$$\text{Hyponym Tree Score} = \frac{\sum_{i=1}^{c\_node} \text{HyponymCount}(w_i)}{c\_node \times \text{tree\_height}} \tag{1}$$

where:

- HyponymCount($w_i$): The number of hyponyms for the $i$-th word in the dependency tree.
- *c_node*: The total number of nodes in the dependency tree.
- tree_height: The maximum depth of the dependency tree.

$$\text{Hyponym Tree Score}_{\text{weighted}} = \frac{\sum_{i=1}^{c_node} \text{HyponymCount}(w_i) \cdot NodeHeight(w_i)}{c\_node \times \text{tree\_height}} \tag{2}$$

where:

- HyponymCount($w_i$): The number of hyponyms for the $i$-th word.
- NodeHeight($w_i$): The height of the $i$-th word in the dependency tree.
- *c_node*: The total number of nodes in the dependency tree.
- tree_height: The maximum depth of the dependency tree.

When considering the implementation options, the dependency tree of a sentence can be created using two popular NLP libraries, namely, Stanza (*Qi et al., 2020*) and SpaCy (*Honnibal et al., 2020*). It has been observed that the dependency tree representation for the same sentence differs between SpaCy and Stanza. Figures 3 and 4 shows the dependency trees created for a sample sentence using Stanza and SpaCy, respectively. The

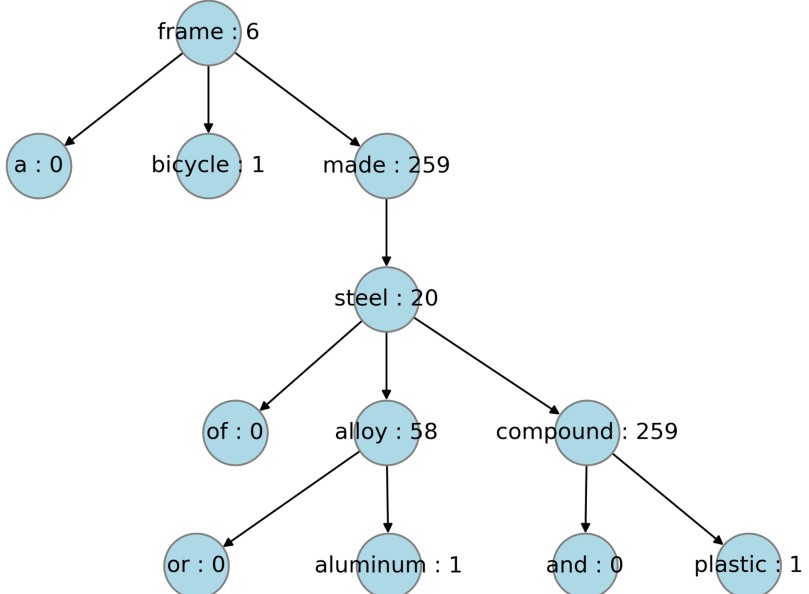

**Figure 3 Hyponym tagged dependency tree with Stanza.**

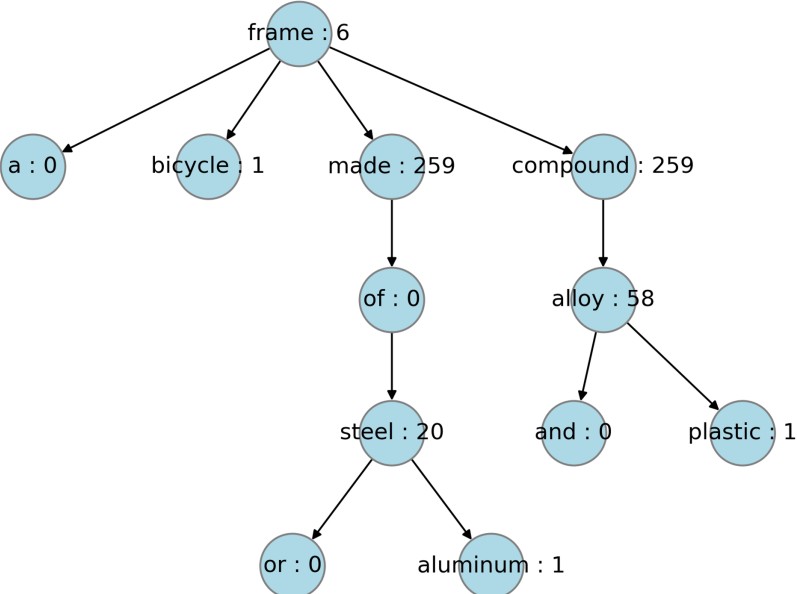

**Figure 4 Hyponym-tagged dependency tree with SpaCy.**

hyponym tree score is dependent upon the dependency tree structure. For this reason, the score calculation was evaluated using both libraries, and a third option was created by averaging the sentence-level scores generated by both libraries. Thus, six candidate hyponym tree scores were generated for further evaluation: three based on SpaCy, Stanza, and averaging, and the weighted versions of all three. Table 4 summarises the HTS candidates generated for evaluation. The difference between the options is primarily based

**Table 4  HTS candidates evaluated.**

| HTS | Description |
| --- | --- |
| hts_spacy | Dependency tree generated using Spacy and nodes are not weighted |
| hts_spacy_wtd | Dependency tree generated using Spacy and nodes are weighted |
| hts_stanza | Dependency tree generated using Stanza and nodes are not weighted |
| hts_stanza_wtd | Dependency tree generated using Stanza and nodes are weighted |
| hts_avg | Sentence level average of hts_spacy and hts_stanza |
| hts_avg_wtd | Sentence level average of hts_spacy_wtd and hts_stanza_wtd |

on two factors: the NLP library used to generate the dependency structure of a sentence and whether the node-level score (hyponym count of that word) is multiplied by a weight. The weight corresponds to the height-based level value, where the leaf node is assigned to level 1, and the root node is assigned level N for a tree with N levels.

### Hyponym tree score validation

Length-based indicators like 'first claim length' blindly treat lengthy claims as specific and short claims as broader, irrespective of the semantics. *Ragot (2023)* presented a set of fixed-length representative claims with varying scopes in section C1 of their work to study the claim scope. The same set of claims is used in this work to study the ability of the newly calculated HTS candidate values. The HTS candidate values are calculated with all the sample claims and presented in Table 5. The sentences are arranged in the descending order of their scope. Figure 5 is a normalized plot of the scope values generated for sample sentences using all the six HTS candidates under evaluation. As per the results, HTS candidates can show scope reduction, whereas the word count fails to represent any scope change. However, due to the close similarity between the results from all the HTS candidates under evaluation, a decision is made to generate all six HTS candidate scores for the entire dataset and to make the final HTS candidate selection only after a complete evaluation with the entire dataset.

### Connecting HTS with litigation risk and claim scope

The preliminary validation of the relationship between HTS and claim scope (CS) is demonstrated in "Hyponym Tree Score Validation". The results indicate that higher HTS values correspond to broader CS. Previous studies (*Merges & Nelson, 1994*; *Arinas, 2012*; *Marco, Sarnoff & Charles, 2019*) have established that broader claim scope increases the likelihood of litigation and legal events. By transitive reasoning, the relationship between HTS and patent litigation probability ($P_{\text{lit}}$) can be considered valid. However, not all patents with broad claim scope result in litigation, as litigation requires legal action to be pursued. This observation may weaken the relationship between HTS and litigation probability.

The evaluation model to study the connection between the HTS and CS is summarized as follows:

• Patents with high HTS are likely to have a broader CS.
• Broader CS increases the probability of litigation ($P_{\text{lit}}$).

**Table 5** HTS candidate values generated for fixed-length sample claims.

| Sentence Ref. | hts_spacy | hts_spacy_wtd | hts_stanza | hts_stanza_wtd | hts_avg | hts_avg_wtd | word_count |
|---|---|---|---|---|---|---|---|
| C1.1 | 304.94 | 1,369.15 | 406.89 | 1,566.00 | 355.92 | 1,467.58 | 25 |
| C1.2 | 279.93 | 1,256.26 | 378.24 | 1,454.97 | 329.09 | 1,355.62 | 25 |
| C1.3 | 144.73 | 976.87 | 260.65 | 1,373.70 | 202.69 | 1,175.29 | 25 |
| C1.4 | 145.38 | 883.62 | 252.33 | 1,240.00 | 198.86 | 1,061.81 | 25 |
| C1.5 | 117.92 | 828.69 | 218.33 | 1,172.00 | 168.13 | 1,000.35 | 25 |
| C1.6 | 87.80 | 664.80 | 187.14 | 1,190.57 | 137.47 | 927.69 | 25 |
| C1.7 | 99.00 | 694.80 | 168.69 | 973.80 | 133.84 | 834.30 | 25 |
| C1.8 | 103.99 | 695.88 | 170.92 | 964.77 | 137.46 | 830.33 | 25 |

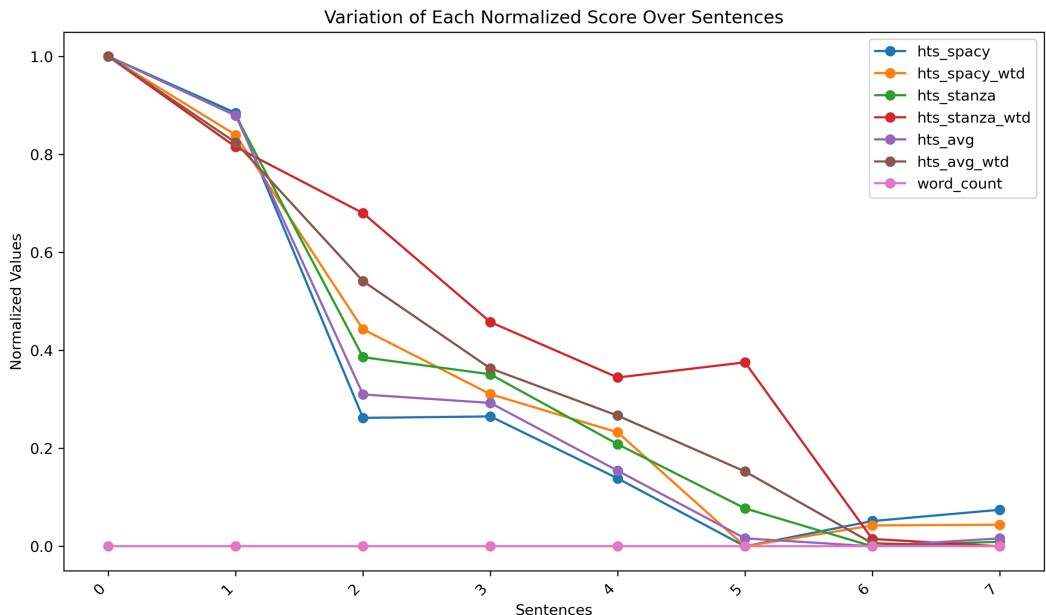

**Figure 5** Claim scope representation using word count and HTS candidates.

- Patents with high HTS and high $P_{\text{lit}}$ are indicative of broader CS.
- Observation: Not all patents with broad Claim Scope (CS) will result in litigation ($P_{\text{lit}}$).

**Objective:** A high HTS strongly predicts $P_{\text{lit}}$, which indicates a broader CS. This provides a scientific basis for using HTS as a quantification method for claim scope and offers a robust framework for patent strategy formulation and risk assessment.

**Predicate logic:**

Let HTS($x$): Patent $x$ has a high Hyponym Tree Score.

Let CS($x$): Patent $x$ has a broad Claim Scope.

Let $P_{\text{lit}}(x)$: Patent $x$ has a high probability of litigation.

**Statements:**

1. $\forall x\,(\mathrm{HTS}(x) \to \mathrm{CS}(x))$: High HTS implies broad CS.
2. $\forall x\,(\mathrm{CS}(x) \to P_{\mathrm{lit}}(x))$: Broad CS implies a high probability of litigation.
3. $\forall x\,((P_{\mathrm{lit}}(x) \wedge \mathrm{HTS}(x)) \to \mathrm{CS}(x))$: High $P_{\mathrm{lit}}$ and HTS imply broad CS.
Observation $\exists x\,(\mathrm{CS}(x) \wedge \neg P_{\mathrm{lit}}(x))$: Not all patents with broad CS result in litigation.

**Proof: Hypothesis:** $\forall x(\mathrm{HTS}(x) \to \mathrm{CS}(x))$

Proof. 1. Assume $\mathrm{HTS}(x)$ for an arbitrary patent $x$.
2. From statement 1, $\mathrm{HTS}(x) \to \mathrm{CS}(x)$, so $\mathrm{CS}(x)$ holds.
3. From statement 2, $\mathrm{CS}(x) \to P_{\mathrm{lit}}(x)$, thus $P_{\mathrm{lit}}(x)$ holds.
4. From statement 3, $(P_{\mathrm{lit}}(x) \wedge \mathrm{HTS}(x)) \to \mathrm{CS}(x)$.
5. Given $P_{\mathrm{lit}}(x)$ and $\mathrm{HTS}(x)$, concludes $\mathrm{CS}(x)$.

Therefore, high HTS implies broad CS.

### Selecting the HTS best candidate

The values of all the HTS candidates are calculated for the entire dataset. Table 6 documents the statistics of the different HTS candidates under evaluation. The distribution of the IPC sections in the dataset is shown in Fig. 1. Section G has the most samples in the dataset. Figure 6 shows the average HTS values for non-litigated and litigated patents belonging to each section. This justifies the Proof "Connecting HTS with Litigation Risk and Claim Scope", on the IPC section level, litigated patents have a higher HTS value than the non-litigated patents and supports the connection between the Litigation probability and HTS value.

The most suitable candidate to represent the CS has to be selected from the six HTS candidates. This section presents seven experiments designed to assess the relative merit of the HTS candidate in litigation prediction. The difference between the experiments is only in the features used for the classification. Set of standard pre-grant features, termed as the baseline features, include 'bwd_cits', 'npl_cits', 'claims_x', 'num_dependent_claims', 'num_independent_claims', 'assignee_pcount', fc_word_count, avg_claim_length and 'num_inventors'. Details of these features are documented in Table 3. Each experiment used random forest, XGBoost, support vector classifier (SVC) and balanced random forest (BRF) models to perform litigation prediction to assess the impact of including the HTS candidate feature with the baseline features.

Figure 7 presents the changes in the accuracy of the litigation prediction during each experiment with different prediction models. Experiment A (Exp-A) performs the prediction by using only the baseline features. Experiments B to G added each HTS candidate along with the baseline features. In all the experiments, XGBoost resulted in the best prediction results. Table 7 presented the features used in each experiment and the best prediction performance achieved with XGBoost. Table 8 shows the correlation between the HTS candidates and other existing patent scope or value indicators. A larger value of HTS indicated a higher litigation probability.

**Table 6 Statistics of all the features.**

| Feature | Count | Mean | Std Dev | Min | 25% | 50% | 75% | Max |
|---|---|---|---|---|---|---|---|---|
| bwd_cits | 81,794 | 26.202 | 70.963 | 0.000 | 5.000 | 11.000 | 24.000 | 6,732.000 |
| npl_cits | 81,794 | 8.950 | 35.334 | 0.000 | 0.000 | 0.000 | 4.000 | 2,128.000 |
| claims_x | 81,794 | 18.313 | 18.481 | 1.000 | 8.000 | 15.000 | 22.000 | 887.000 |
| avg_claim_length | 81,794 | 41.688 | 33.095 | 1.000 | 23.762 | 33.714 | 49.000 | 3,198.000 |
| num_dependent_claims | 81,794 | 15.043 | 16.933 | 0.000 | 6.000 | 12.000 | 19.000 | 886.000 |
| num_independent_claims | 81,794 | 3.326 | 3.365 | 0.000 | 2.000 | 3.000 | 4.000 | 155.000 |
| assignee_pcount | 81,794 | 7,246.173 | 21,565.206 | 1.000 | 15.000 | 176.000 | 2,656.000 | 156,703.000 |
| num_inventors | 81,794 | 2.378 | 1.749 | 1.000 | 1.000 | 2.000 | 3.000 | 31.000 |
| fc_word_count | 81,794 | 167.448 | 114.100 | 2.000 | 100.250 | 147.000 | 209.000 | 7,711.000 |
| hts_spacy | 81,794 | 1,356.222 | 1,621.952 | 1.000 | 470.201 | 923.674 | 1,683.081 | 62,917.851 |
| hts_spacy_wtd | 81,794 | 10,839.710 | 14,770.509 | 1.500 | 3,320.736 | 6,830.528 | 13,207.373 | 663,865.829 |
| hts_stanza | 81,794 | 1,418.662 | 1,824.045 | 0.000 | 486.020 | 951.797 | 1,735.294 | 73,828.074 |
| hts_stanza_wtd | 81,794 | 10,250.467 | 13,744.568 | 0.000 | 3,191.651 | 6,489.478 | 12,493.479 | 650,009.383 |
| hts_avg | 81,794 | 1,387.442 | 1,668.136 | 0.700 | 485.400 | 943.672 | 1,715.445 | 68,372.963 |
| hts_avg_wtd | 81,794 | 10,545.088 | 14,217.464 | 1.500 | 3,267.236 | 6,674.460 | 12,860.270 | 656,937.606 |

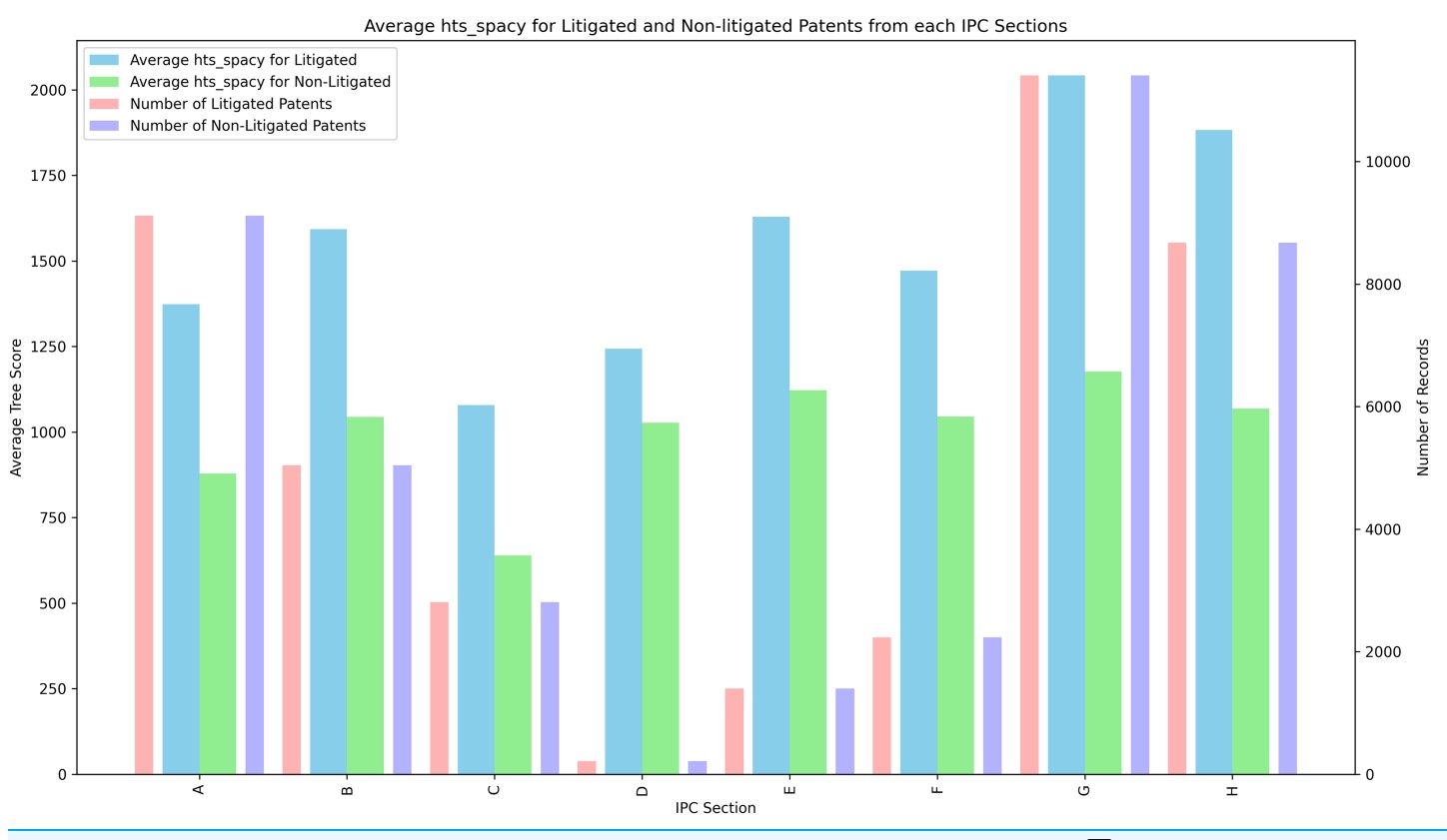

**Figure 6 Average HTS for each IPC section.**

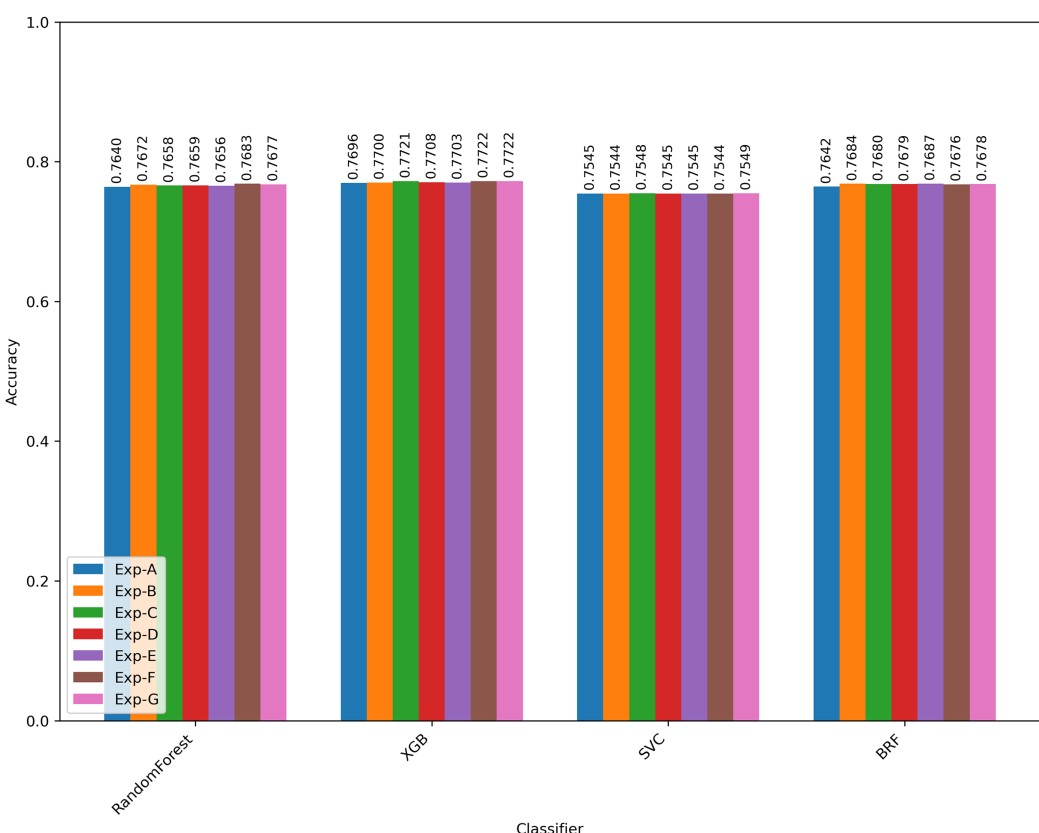

**Figure 7 Litigation prediction accuracy for all ML models during each experiment.**

**Table 7 Performance metrics for experiments using XGBoost.**

| Experiment | Features | Accuracy | Precision | Recall | F1-score | AUC |
|---|---|---|---|---|---|---|
| Exp-A | Baseline features | 0.770 | 0.788 | 0.738 | 0.762 | 0.770 |
| Exp-B | Baseline features + hts_spacy | 0.770 | 0.785 | 0.743 | 0.764 | 0.770 |
| Exp-C | Baseline features + hts_spacy_wtd | 0.772 | 0.789 | 0.743 | 0.765 | 0.772 |
| Exp-D | Baseline features + hts_stanza | 0.771 | 0.787 | 0.742 | 0.764 | 0.771 |
| Exp-E | Baseline features + hts_stanza_wtd | 0.770 | 0.786 | 0.743 | 0.764 | 0.770 |
| Exp-F | Baseline features + hts_avg | 0.772 | 0.787 | 0.746 | 0.766 | 0.772 |
| Exp-G | Baseline features + hts_avg_wtd | 0.772 | 0.789 | 0.744 | 0.766 | 0.772 |

Figure 8 shows the variation of the metric from the experiment A results during each experiment. Accordingly, prediction accuracy improved marginally with the introduction of the HTS candidate features. Whenever the positive correlation between the HTS and Litigation probability is valid, the positive correlation between the HTS and Claim Scope is also valid as per the proof "Connecting HTS with Litigation Risk and Claim Scope". Thus, the relationship between the HTS and Patent Scope is reconfirmed.

From the evaluation results, hts_stanza and hts_stanza_wtd are unambiguously ruled out. The hts_avg_wtd produced slightly better prediction results compared to hts_spacy

**Table 8 Correlation between the HTS candidates and other indicators.**

| Feature | fwd_cits5 | PQI6 | family_size | grant_lag | fc_word_count | litigation_label |
|---|---|---|---|---|---|---|
| hts_spacy | 0.092 | 0.328 | 0.007 | 0.116 | 0.119 | 0.205 |
| hts_spacy_wtd | 0.071 | 0.241 | −0.033 | 0.103 | 0.192 | 0.152 |
| hts_stanza | 0.085 | 0.300 | 0.020 | 0.105 | 0.125 | 0.185 |
| hts_stanza_wtd | 0.068 | 0.240 | −0.025 | 0.099 | 0.201 | 0.148 |
| hts_avg | 0.091 | 0.323 | 0.014 | 0.114 | 0.126 | 0.201 |
| hts_avg_wtd | 0.070 | 0.241 | −0.029 | 0.101 | 0.197 | 0.150 |

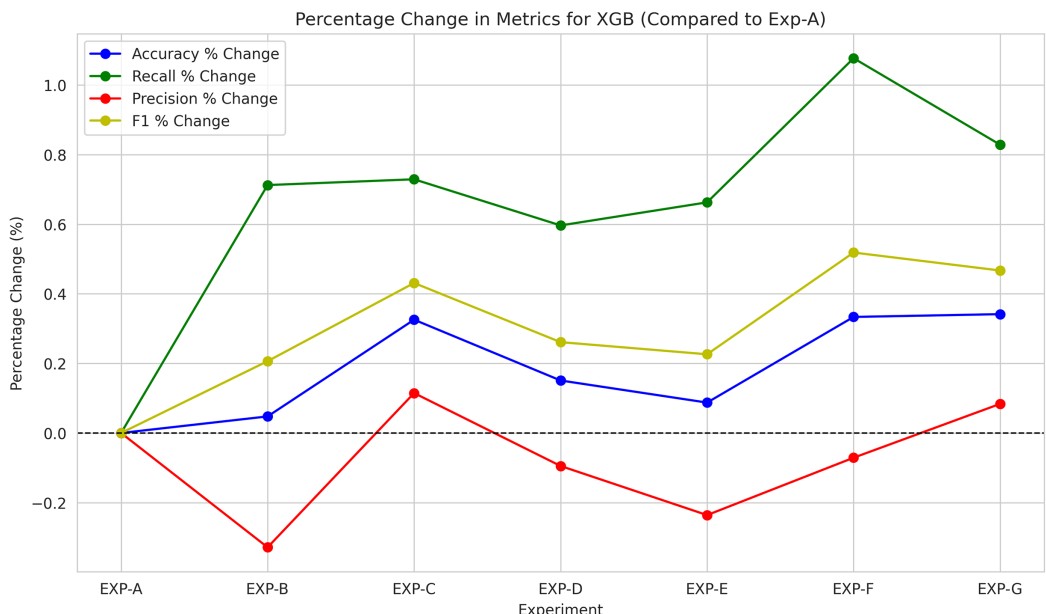

**Figure 8 Percentage variation of XGBoost prediction performance compared to Exp-A.**

and hts_spacy_wtd. During the experiments, it was observed that the creation of a stanza-based dependency graph failed for several sentences. Calculating the average scores requires both stanza and spacy-based dependency tree creation. The average scores were also discarded to avoid the dependency tree creation issues observed with Stanza. The remaining candidates are hts_spacy and hts_spacy_wtd, and experiments B and C evaluate the prediction with these features. When comparing the results between experiments B and C, hts_spacy_wtd produces slightly better results. Figure 9 represents the information gain of the candidate HTS features. As per the information gain of the candidates, hts_spacy shall be the candidate.

To overcome the ambiguity, a feature-based extremes evaluation was conducted to investigate the relationship between the candidate features and the litigation label and the results are presented in Fig. 10. Specifically, the top 100 and bottom 100 records based on the candidate feature's values were identified, and their litigation labels were analyzed. The objective was to determine whether the top 100 records predominantly correspond to the

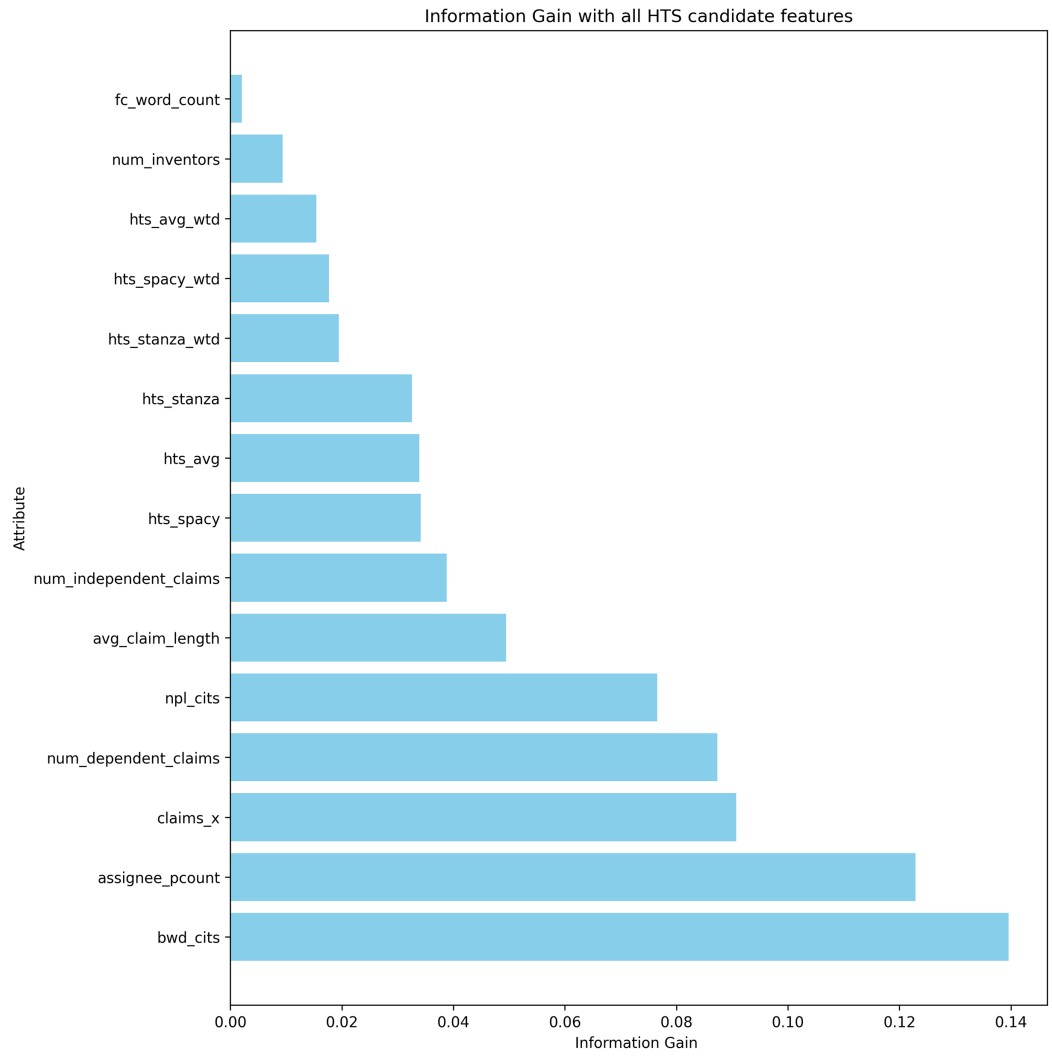

**Figure 9 Information gain of the features used in litigation prediction.**

positive class (litigated) and the bottom 100 records to the negative class (non-litigated). This analysis provided insights into the HTS candidate feature's discriminative power, thereby offering a supplementary validation of the feature's relevance in the litigation prediction task. Extremes analysis and correlation analysis resulted in favour of hts_spacy against hts_spacy_wtd. In these circumstances, hts_spacy was selected as the final candidate for the HTS. All further references to HTS will indicate the usage of hts_spacy as the indicator for the claim scope representation.

## Part 2: litigation prediction model development

The primary objective of this phase was to develop a litigation prediction model that incorporates claim scope understanding while remaining independent of post-grant features, thereby ensuring its applicability to patent drafts. Transformer-based, pre-trained
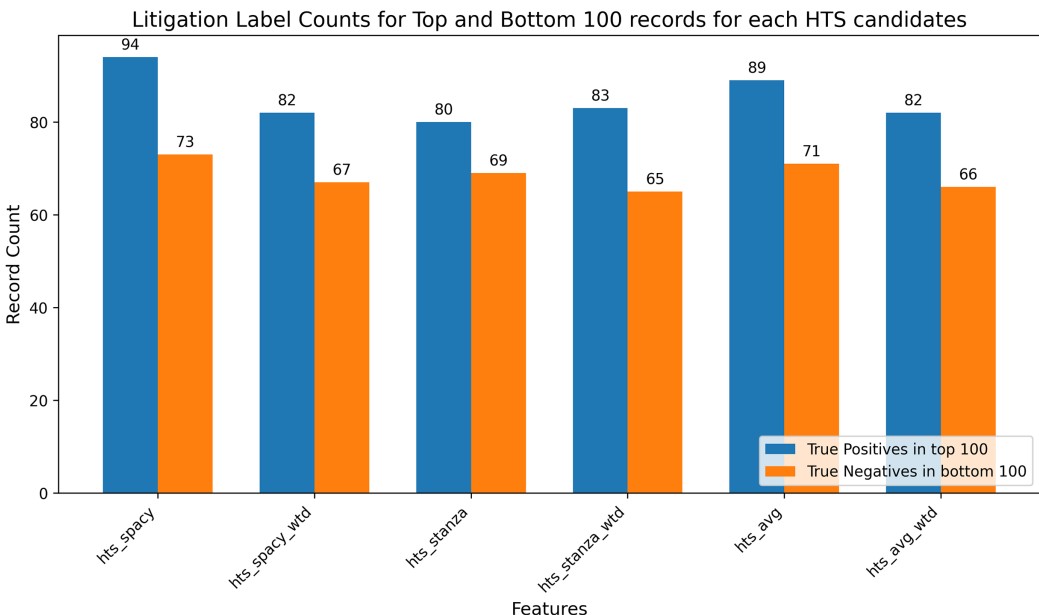

**Figure 10 Extremes analysis for predictive label validation.**

models have garnered considerable attention for text-based analyses, owing to their exceptional ability to capture contextual nuances (*Gasparetto et al., 2022*). Among these, BERT models are widely recognized for their effectiveness in language comprehension tasks, making them a suitable choice for litigation prediction using claim text. Initial experiments employing BERT demonstrated superior predictive performance compared to traditional machine learning models, validating the effectiveness of transformer-based approaches in this domain.

The results obtained using the BERT (bert-base-uncased) model were based on processing only the claim text, limited to the first 512 words, which may result in incomplete comprehension of lengthy patent claims. In this context, instead of the usual chunking approach, a multifeature fusion approach was devised, integrating both claim text and numerical features to enhance prediction accuracy. "Part 1: Claim Scope Indicator Development" identified HTS as a potential indicator of claim scope, with the hts_spacy variant chosen as the optimal feature. To augment the scope-awareness of the model, HTS was incorporated alongside the baseline numerical features described in "Selecting the HTS Best Candidate". As a result, a Multifeature Fusion Deep Learning Model was proposed and evaluated, leveraging both textual and numerical modalities to predict litigation risks effectively.

### Proposed litigation prediction model

The proposed model, referred to as MAPRA (Multifeature BERT-Powered Fusion for Author-level Patent Litigation Risk Analysis), is specifically designed to assess litigation risks in patent drafts. The model's architecture aims to capture both the semantic

intricacies of claim text and the scope awareness conveyed by numerical features. The textual component is processed through a BERT-based encoder, which extracts semantic information essential for identifying litigation-prone claims. Concurrently, the numerical features provide supplementary insights, such as claim scope and other litigation-relevant factors, resulting in a holistic view of the data. Additionally, author and assignee details, often absent in the claim text, are incorporated as numerical features to enhance prediction accuracy. The ten numeric features used in this analysis are *bwd_cits*, *npl_cits*, *claims_x*, *avg_claim_length*, *num_dependent_claims*, *num_independent_claims*, *assignee_pcount*, *num_inventors*, *hts_spacy*, and *fc_word_count*.

Data preprocessing plays a pivotal role in ensuring the robustness of the model. Textual claims are tokenized and encoded using the BERT tokenizer, which standardizes the input by padding sequences to a fixed length. This ensures compatibility with the BERT encoder while preserving consistency in input dimensions. Simultaneously, numerical features are imputed, trimmed, normalized, and scaled to maintain uniformity across the dataset. Despite the 2% real-world prevalence of the positive class, coverage of positive samples is ensured through a 10% oversampling scheme, thereby guaranteeing that positive examples are included in each training epoch. The processed data are stratified into training (80%), validation (10%), and test (10%) sets, with both balanced (1:1) and realistic imbalanced ($\approx 2\%$ positives) splits created for evaluation. This separation is critical to mitigating overfitting and validating the model's generalizability to unseen data.

### Model architecture and workflow

The proposed MAPRA model integrates textual and numerical data modalities to predict a binary litigation outcome (litigated/not litigated). The MAPRA architecture is specifically designed to assess litigation risks, combining semantic intricacies from claim text with numeric indicators representing claim scope, inventor, and assignee attributes. Figure 11 illustrates the complete workflow of the model.

1. **Input representation**

Let $x$ represent tokenized patent claim text and $\mathbf{n} = (n_1, n_2, \ldots, n_{10})$ represent the vector of associated numeric features.

2. **Tokenization and embedding**

   ○ **Tokenization:** Claim texts are tokenized and padded to length 512 offline using BERT tokenizer, yielding cached *input_ids* and *attention_mask*.

   ○ **BERT embedding:** Text embeddings are generated by fine-tuning layers 8–11 of BERT while keeping layers 0–7 frozen. The output embeddings are:

   $$\mathbf{E} = (\mathbf{e}_1, \ldots, \mathbf{e}_T), \quad \mathbf{e}_i \in \mathbb{R}^d, d = 768.$$

3. **CLS token representation**

The embedding of the [CLS] token is extracted as a semantic representation of the claim:

$$\mathbf{e}_{\text{CLS}} = \mathbf{E}_{\text{CLS}} \in \mathbb{R}^{768}.$$

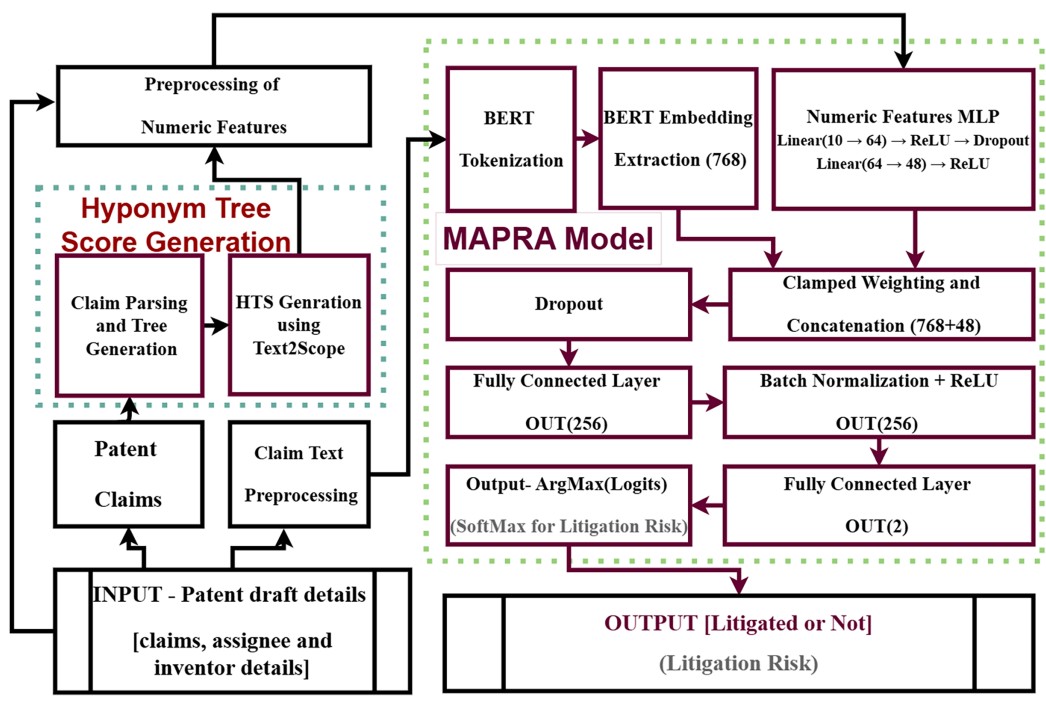

**Figure 11 Litigation prediction using MAPRA model.**

4. **Combining text and numeric features**

- Numeric features are subjected to median imputation, trimming, and Min-Max normalization to the range $[0, 1]$.

- Numeric embeddings ($\mathbf{e}_{\text{Num}} \in \mathbb{R}^{48}$) are generated through a multilayer perceptron (MLP):

  $$\text{Linear}(10 \to 64) \to \text{ReLU} \to \text{Dropout}(0.4) \to \text{Linear}(64 \to 48) \to \text{ReLU}.$$

- A learnable modality weighting vector $\mathbf{p} \in \mathbb{R}^2$, clamped and normalized *via* softmax, yields weights $w_0, w_1$, resulting in:

  $$\mathbf{f} = [w_0 \mathbf{e}_{\text{CLS}}; \; w_1 \mathbf{e}_{\text{Num}}] \in \mathbb{R}^{816}.$$

5. **Classification layer**

- The combined feature vector $\mathbf{f}$ is passed through a feedforward neural network.

- Dropout with probability 0.33 is used.

- Fully connected layer, batch normalization (BN), and ReLU activation:

  $$\mathbf{h}_1 = \text{ReLU}(\text{BN}(\mathbf{W}_1 \mathbf{f} + \mathbf{b}_1)), \quad \mathbf{W}_1 \in \mathbb{R}^{256 \times 816}.$$

- The logits for binary classification are computed as:

  $$\mathbf{z} = \mathbf{W}_2 \mathbf{h}_1 + \mathbf{b}_2, \quad \mathbf{W}_2 \in \mathbb{R}^{2 \times 256}.$$

○ The softmax function is applied to obtain the predicted probabilities:

$$\widehat{\mathbf{y}} = \mathrm{softmax}(\mathbf{z}) \in \mathbb{R}^2.$$

where $\widehat{\mathbf{y}} \in \mathbb{R}^2$ represents the probabilities for the two classes.

6. **Loss function**

To effectively handle class imbalance during training, the MAPRA model employs a cost-sensitive variant of the binary cross-entropy loss known as the *focal loss*. The standard binary cross-entropy (CE) loss for binary classification, with true label $y \in \{0, 1\}$ and predicted probability $\hat{p}$, is given by:

$$\mathscr{L}_{\mathrm{CE}} = -[y \log(\hat{p}) + (1 - y) \log(1 - \hat{p})].$$

The focal loss extends this by emphasizing difficult-to-classify examples, and is defined as:

$$\mathscr{L}_{\mathrm{FL}} = \alpha(1 - p_t)^\gamma \, \mathscr{L}_{\mathrm{CE}},$$

where $p_t$ is the model's predicted probability for the true class, $\gamma$ is the focusing parameter (set to 2), and $\alpha$ balances the class weights (set to 1). To further account for severe class imbalance, class-specific weights are employed as follows:

$$w_+ = \sqrt{\frac{1 - \pi_+}{\pi_+}}, \quad w_- = 1, \quad \pi_+ = 0.02.$$

Thus, the final weighted focal loss for the dataset of size $N$ is expressed as:

$$\mathscr{L} = -\frac{1}{N} \sum_{i=1}^{N} w_{y_i} [\alpha(1 - p_{t,i})^\gamma (y_i \log(\hat{p}_i) + (1 - y_i) \log(1 - \hat{p}_i))].$$

This combined loss function enhances the model's sensitivity toward the minority class, significantly improving recall performance on rare litigated patents.

7. **Training objective**

○ The model parameters $\Theta$ are optimized by minimizing the focal loss ($\mathscr{L}_{FL}$) across the training dataset using the AdamW optimizer (learning rate $1.2 \times 10^{-5}$, weight decay 0.01), cosine scheduler with a 5% warmup phase, gradient clipping (maximum norm of 1.0), and early stopping based on validation area under the precision-recall curve (AUPRC):

$$\min_{\Theta} \frac{1}{N} \sum_{i=1}^{N} \mathscr{L}_{FL}(y_i, \widehat{\mathbf{y}}_i),$$

where $N$ denotes the total number of training samples, $y_i$ is the true label, and $\widehat{\mathbf{y}}_i$ is the predicted probability vector for the $i$-th training example.

8. **Evaluation under true class imbalance**

Given the true positive prevalence

$$\pi_+ = \frac{N_+}{N} = 0.02,$$

where $N_+$ is the number of litigated patents in a test set of size $N$, the classifier produces scores $\hat{p}_i = P(y_i = 1 \mid x_i)$. Instead of relying solely on the default threshold 0.5, the decision threshold is calibrated using multiple criteria on a 2%-positive validation set:

○ **F1-optimal threshold:**

$$t_{F1} = \arg\max_t F_1(t), \quad F_1(t) = \frac{2P(t)\,R(t)}{P(t) + R(t)},$$

where precision ($P$) and recall ($R$) at threshold $t$ are defined as:

$$P(t) = \frac{\sum_{i=1}^N \mathbb{I}[\hat{p}_i \geq t]\,y_i}{\sum_{i=1}^N \mathbb{I}[\hat{p}_i \geq t]}, \quad R(t) = \frac{\sum_{i=1}^N \mathbb{I}[\hat{p}_i \geq t]\,y_i}{\sum_{i=1}^N y_i}.$$

○ **Accuracy-optimal threshold**:

$$t_{acc} = \arg\max_t Acc(t).$$

○ **Fixed 2% flag-rate threshold ($98^{th}$ percentile)**:

$t_{2\%}$ such that $\mathbb{P}[\hat{p}_i \geq t_{2\%}] = 0.02$.

Ranking quality under extreme class imbalance is further evaluated using Precision@K and Recall@K:

$$P@K = \frac{1}{K} \sum_{i=1}^K y_{(i)}, \quad R@K = \frac{1}{\sum_i y_i} \sum_{i=1}^K y_{(i)},$$

where $y_{(i)}$ denotes the ground-truth label of the $i$-th highest-scoring example. Additionally, the Area Under the Precision–Recall Curve (AUPRC),

$$AUPRC = \int_0^1 P(R)\,dR,$$

is monitored and optimized, given its superior informativeness over area under the receiver operating characteristic curve (ROC-AUC) in highly imbalanced contexts ($\pi_+ \ll 0.5$).

The model training explicitly uses a class-weighted focal loss:

$$\mathscr{L}_{FL} = -\frac{1}{N} \sum_{i=1}^N w_{y_i} \alpha (1 - p_{t,i})^\gamma [y_i \log \hat{p}_i + (1 - y_i) \log(1 - \hat{p}_i)],$$

with class weights

$$w_+ = \sqrt{\frac{1 - \pi_+}{\pi_+}} \approx 7, \quad w_- = 1, \quad \alpha = 1, \quad \gamma = 2.$$

These class weights imply that false negatives incur significantly higher penalties due to the extreme class imbalance ($\pi_+ = 0.02$), effectively enhancing recall for rare litigation-positive cases. Combined with offline positive-class oversampling (10%), comprehensive numeric preprocessing (imputation, trimming, scaling), multi-threshold calibration, and rigorous ranking metrics, this approach effectively maximizes recall for rare litigation events while controlling false-positive predictions.

9. **Model inference**

○ During inference, the class label is predicted by selecting the class with the highest probability:

$$\hat{y} = \arg\max_{c} \widehat{\mathbf{y}}_c.$$

Optionally, calibrated thresholds (*e.g.*, F1-optimal, accuracy-optimal, or fixed flag-rate thresholds) may be applied to enhance inference quality under class imbalance.

**Summary of the workflow**

1. Offline tokenize and pad claim texts (length 512) using the BERT tokenizer.
2. Extract the [CLS] token embedding from fine-tuned BERT as the textual representation.
3. Preprocess numeric features (imputation, trimming, scaling) and encode them using a dedicated numeric MLP.
4. Combine textual and numeric embeddings using learnable modality clamped-weighting and concatenation.
5. Pass the weighted combined embedding through a feedforward neural network with dropout and batch normalization.
6. Compute class probabilities using softmax and minimize the focal loss during training.
7. Calibrate optimal decision thresholds on a validation set.
8. Predict class labels based on calibrated probabilities during inference.

**Training and evaluation**

Training involves minimizing the focal loss using the AdamW optimizer, gradient clipping, dropout regularization, and early stopping based on validation performance (AUPRC). Final evaluation on balanced and realistic imbalanced splits involves comprehensive metric computation and visualization of receiver operating characteristic curves, precision-recall curves, and confusion matrices. This architecture, which combines NLP techniques with numeric feature integration, provides a robust framework for predicting litigation risk in legal analytics.

# RESULTS

The first part of the work developed an indicator named 'Hyponym Tree Score' for patent claim text scope quantification. To generate the proposed score from the patent claim text input, algorithms were designed and implemented using two popular NLP libraries, Spacy and Stanza. Thus, six candidate hyponym scores were generated for further evaluation. The validity of the HTS candidates was initially evaluated using the sample claims to verify their ability to distinguish the scope variation from a set of fixed-length claims. Based on the positive results from that experiment, seven experiments were conducted to perform the litigation prediction; the results were consolidated in Table 7. In addition to the

**Table 9 Performance metrics for BERT-based models.**

| Experiment | Features | Accuracy | Precision | Recall | F1-score | AUC |
|---|---|---|---|---|---|---|
| BERT | Only claim text | 0.8099 | 0.8099 | 0.8099 | 0.8099 | 0.8099 |
| MAPRA | Claim text + Features from Exp-B | 0.8005 | 0.7779 | 0.8411 | 0.8083 | 0.8776 |

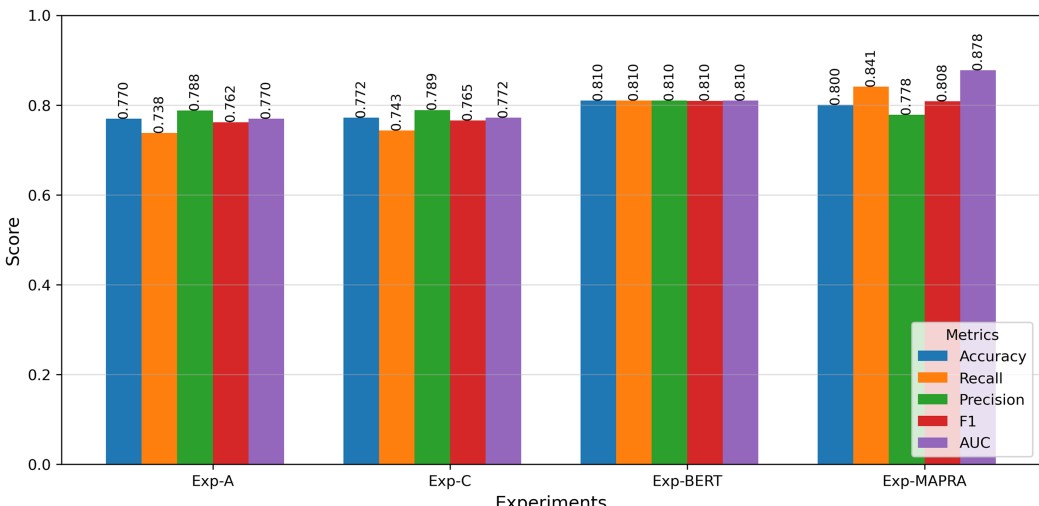

**Figure 12 Comparison of the performance metrics with different experiments.**

prediction results, extremes analysis, correlation and information gain results are also considered to identify the final candidate for HTS. Based on observations, the non-weighted HTS generated using the Spacy library was selected as the final candidate for the proposed HTS to quantify the patent claim scope.

The second part of this work aimed to develop a litigation prediction model that relies solely on early-stage features, ensuring its applicability to both patent drafts and granted patents. The key direction adopted in the development of the proposed 'MAPRA' model was utilising the BERT Model for claim text understanding and augmenting the text information with additional numerical features by designing a Multifeature Fusion approach. Among the BERT options, the BERT base (bert-base-uncased) model was used for text understanding purposes. As a preliminary step, a baseline BERT model using only claim text was implemented to validate its capability in litigation prediction. The results were comparable to those achieved in the first part of the study. Building on this validation, the MAPRA model was developed and tested using both the claim text and the ten numerical features from Part 1, Experiment B. Table 9 presents the prediction metrics for the BERT-based experiments. Figure 12 shows the accuracy improvement in litigation prediction with different models.

To enhance model interpretability and build confidence in its predictive behavior, SHAP (SHapley Additive exPlanations) values were employed to quantify the contribution of individual input features to the model's output. Figure 13 presents a SHAP summary

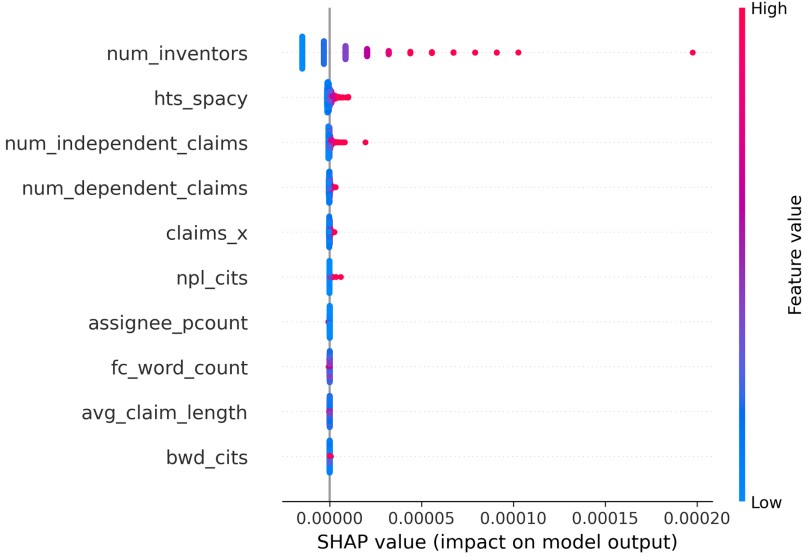

**Figure 13 SHAP summary plot showing feature importance based on average impact on model output.**

plot ranking features based on their impact on model predictions. The most influential inputs were *num_inventors*, *hts_spacy*, and *num_independent_claims*. Notably, *num_inventors*, despite its high ranking, represents external metadata unrelated to the content of patent claims. In contrast, *hts_spacy*, the proposed semantic scope indicator derived from claim text, was the most impactful claim-related feature. Its high SHAP values indicate that the semantic structure of claims plays a substantial role in predicting litigation risk. These results validate the relevance of traditional bibliometric indicators while empirically demonstrating the added value of incorporating HTS, the proposed claim scope indicator. Overall, the findings reinforce the interpretability of the MAPRA model and underscore the potential of *hts_spacy* as a meaningful early-stage indicator of litigation risk.

The primary objective of the proposed model is to serve as an early-stage risk assessment tool during the patent drafting process. In this context, the model functions as a screening mechanism, where false negatives (*i.e.*, failing to identify potentially litigated patents) pose greater strategic risk than false positives. As such, achieving high recall is essential. The MAPRA model demonstrates superior recall and ROC-AUC compared to all baseline configurations when evaluated using the F1-optimal threshold, which yields its best overall performance and underscores its effectiveness in minimizing missed high-risk cases. These results support its viability as a decision-support tool capable of providing meaningful, actionable insights to patent authors at the draft stage. These early-stage insights enable authors to manage the legal scope of their claims and assess the potential litigation risk of their patents. Figure 14 illustrates improvements in key prediction metrics across different experiments. Compared to the results from Experiment A, the MAPRA model achieved a 4% improvement in prediction accuracy and a 14% improvement in recall. As a pioneering effort in predicting litigation risk for patent drafts, MAPRA cannot

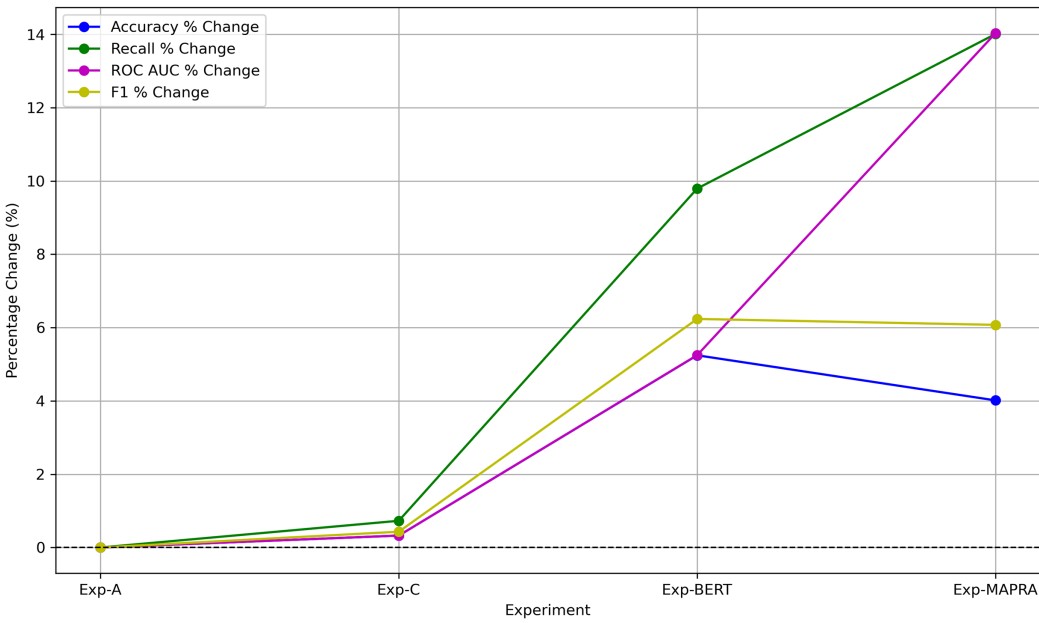

**Figure 14 Percentage improvement in key prediction metrics relative to Exp-A.**

be directly compared to prior models, as no published work to date has addressed this problem at the draft stage. The most comparable existing study (*Juranek & Otneim, 2024*) focuses on litigation prediction using features available immediately after patent grant and reports an AUC of 0.822. In contrast, the MAPRA model achieves a higher AUC of 0.878 while relying exclusively on early-stage features. A key advantage of MAPRA is that it does not depend on post-grant or acquired information, yet it demonstrates superior predictive performance. These results highlight MAPRA's capability to assess litigation risk effectively at both the draft and post-grant stages. To the best of the authors' knowledge, this represents the first published approach explicitly designed for litigation risk prediction during the patent drafting stage.

Although the model was trained on a balanced dataset, its evaluation on a realistically imbalanced test set, comprising approximately 2% litigated and 98% non-litigated patents, demonstrates its effectiveness in identifying high-risk cases. The test set contains 4,173 samples, including only 83 litigated instances. The model achieves a recall of 85.54%, successfully capturing the majority of truly litigated patents. Despite the expected trade-off in such imbalanced settings, it attains a precision of 6.74% and an F1-score of 0.1250. Notably, its Precision@200 is 16%, representing an eightfold improvement over random selection. Additional performance metrics include an accuracy of 76.18%, a ROC-AUC of 0.8786, and an Average Precision (AP) of 0.1909, highlighting the model's strong ranking performance. These results suggest that the model is well suited for prioritization tasks in large-scale patent portfolios (*Saito & Rehmsmeier, 2015*). Moreover, its performance is expected to improve further when trained on a larger dataset that reflects the true class distribution (*Buda, Maki & Mazurowski, 2018*).

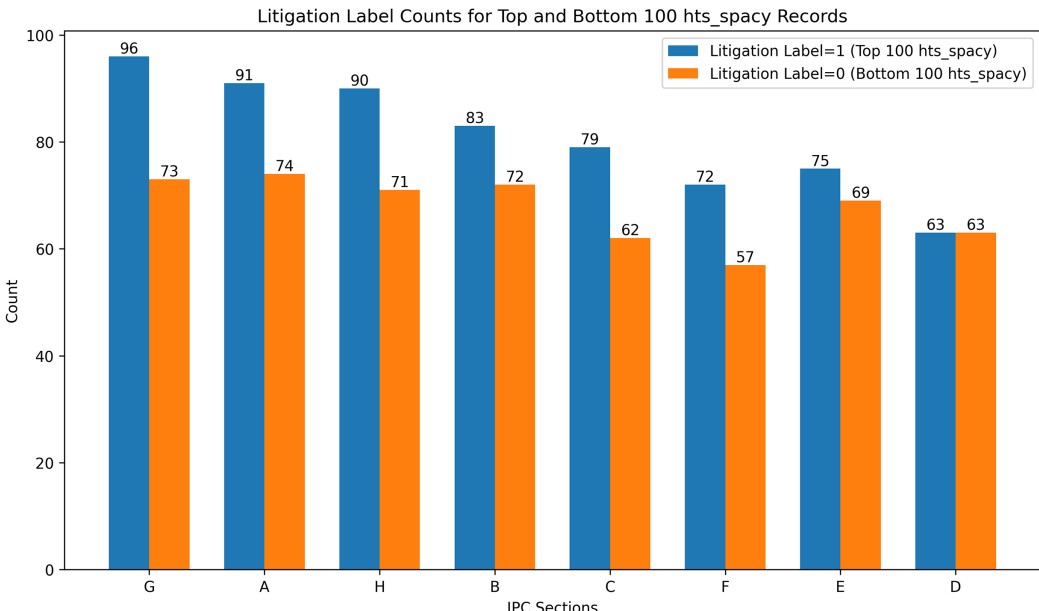

**Figure 15 Litigation label counts for top and bottom 100 hts_spacy records.**

## DISCUSSION

The importance of the patent claim scope and its significance to different stakeholders triggered the study. Prominent indicators used to represent the patent scope are studied in this work. The first part of this work addresses the identified research gap regarding the underutilization of patent claim text semantics in assessing patent scope by proposing HTS, a new claim scope indicator. The impact of including HTS in litigation prediction using different machine learning models was evaluated to identify the most suitable candidate for HTS.

As depicted in Fig. 8, the relative performance improvement of each experiment compared to the baseline experiment is smaller. This is not an unexpected situation because scope capturing is based on identifying the hyponyms for each word. The quality of HTS depends on two factors: how the dependency graph for the sentence is generated and how the hyponym counts for the words in the claim sentences are identified. To generate the dependency tree, the Stanza and Spacy libraries were evaluated, and it was observed that Spacy produces overall better results. The hyponym count is calculated using WordNet, but upon analysis, it was observed that WordNet cannot provide the hyponyms of techno-legal terms, which are key constituents of the patent claim text. It cannot resolve the context-related ambiguity; for example, the word 'tree' in computer science refers to a data structure, whereas in the context of environmental science, it refers to a natural tree. Figure 15 presents the fact that the ability of the HTS to predict the litigation risk potentially varies with the patent's IPC section, which internally refers to the linguistic diversity supported by the hyponym counting mechanism. For WordNet, only the presence or absence of the word is checked, ignoring its context. Another observation is
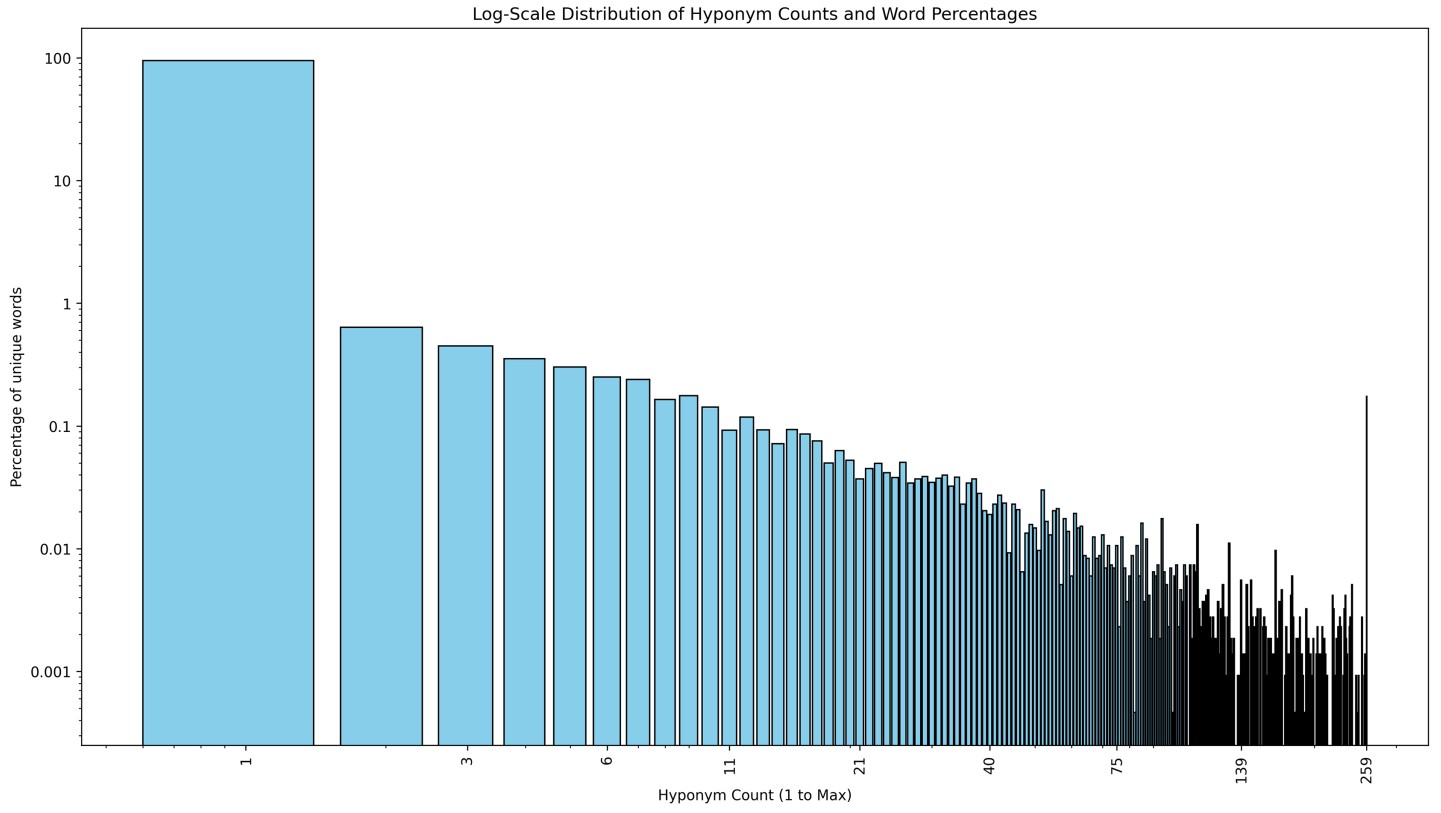

**Figure 16 Log-scale distribution of hyponym counts and word percentages.**

that for certain words, WordNet provides hundreds of hyponyms. This can potentially nullify the significance of all other words in the sentence. To avoid such outliers, in this work, the maximum value of any word's hyponym count is limited to 259, which corresponds to the hyponym count for the top 98.5 percentile. When a word is not a stopword, and there is no hyponym for that word, the original word is considered, and a minimum hyponym count of one is assigned. Figure 16 shows the hyponym count plotted against the percentage of unique words or tokens present in the patent claim text *corpus* of the dataset used in this work. As indicated by this, it is clear that no hyponyms exist in WordNet for most of the words extracted from the patent claims. Although WordNet is the most popular hyponym *corpus*, HTS calculation requires a new *corpus* that includes all scientific, legal, and technical terms to yield better results. Currently, there is no WordNet replacement with context awareness and inclusion of domain-specific terms available in the public domain. To achieve the full potential of the HTS, the development of a new hyponym *corpus* and the re-computation of HTS values using this new *corpus* are recommended. This pioneering study aims to trigger researchers' interest in quantifying the claim text scope based on hyponym count and sentence structure.

This study acknowledges that the connection between patent scope, value, and litigation probability has been reconfirmed. Prior studies have documented the correlation between patent scope and its value and the connection between scope and litigation tendency. No

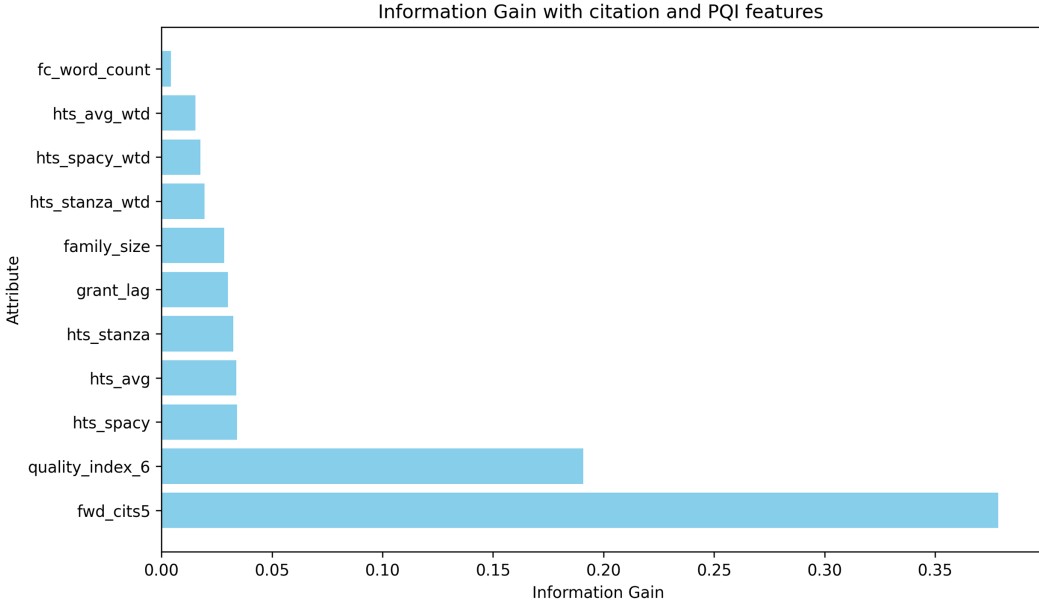

**Figure 17 Information gain of post-grant features and HTS candidates.**

attempts were made in this study to promote or encourage patents with a very broad scope due to the well-known conflicting views on such patents (*Kitch, 1977*; *Klemperer, 1990*; *Gilbert & Shapiro, 1990*; *Merges & Nelson, 1994*; *Chang, 1995*). Such patents are often used as tools to suppress competition. The focus is solely on quantifying patent scope and enabling patent drafters to determine whether the articulated claim scope is higher or lower.

The proposed MAPRA model is specifically designed for litigation prediction at the patent drafting stage. Unlike the prior work focused on granted patents, such as (*Juranek & Otneim, 2024*), which reported an AUC of 0.822, the MAPRA model achieves a higher AUC of 0.8776 while using only pre-grant features. This result demonstrates that strong predictive performance can be achieved without relying on post-grant event data. The use of exclusively draft-stage information makes MAPRA well-suited for early-stage litigation risk assessment. To demonstrate the significance of post-grant features in litigation prediction, Fig. 17 shows the information gain of popular post-grant features such as PQI6, forward citation, grant lag, and family size. In this combination, the contribution of post-grant features is very high. Additionally, it is important to note that HTS candidate features outperform two well-known post-grant features: family size and grant lag. This reconfirms the significance of HTS in litigation prediction for early-stage documents for which the post-grant features are unavailable. Considering that the performance difference between the closest litigation prediction work is just 0.3%, the MAPRA model can also be effectively used for granted patents. By using HTS for claim scope identification and MAPRA model for litigation prediction, patent authors can iteratively modify the patent claim text to obtain an optimal claim scope that balances higher value and improved grant probability.

| Table 10 Performance on imbalanced test data with varying training ratios. | | | | | |
|---|---|---|---|---|---|
| Training ratio | Precision | Recall | F1-score | AP | P@200 |
| 1:1 | 0.0533 | 0.8500 | 0.1003 | 0.2116 | 0.080 |
| 1:2 | 0.0701 | 0.8250 | 0.1292 | 0.2348 | 0.135 |
| 1:3 | 0.0817 | 0.7049 | 0.1465 | 0.1842 | 0.135 |

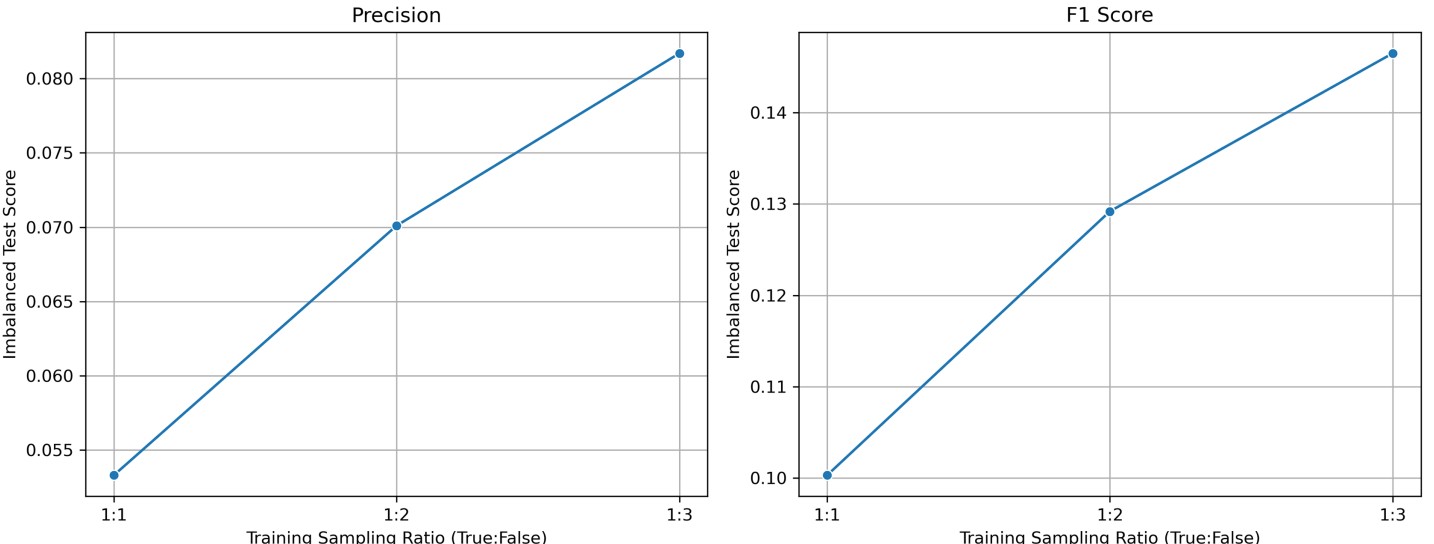

**Figure 18 Impact of training sampling ratios on precision and F1-score with an Imbalanced (2:98) test set.**

To assess the model's real-world applicability, further analysis was conducted on an imbalanced test set reflecting the true distribution of litigated patents (approximately 2% positive cases). While the baseline results, reported earlier, indicate strong performance in terms of recall and ROC-AUC, precision and F1-score were comparatively lower. This behavior is expected in rare-event settings, where even a small number of false positives can significantly affect threshold-sensitive metrics like precision and F1. To better understand the impact of training class distribution on model generalization under deployment-like conditions, additional experiments were conducted using three training configurations, each incorporating 10,000 litigated (positive class) patents. The models were trained with positive-to-negative sampling ratios of 1:1, 1:2, and 1:3, respectively. Each model was evaluated on the 2:98 imbalanced test set. The results, presented in Table 10, show that as the training distribution progressively approximates the true class imbalance, the model exhibits notable gains in several key metrics, including precision, F1-score, and average precision (AP). Precision@200 improves from 0.080 (1:1) to 0.135 (1:2 and 1:3), suggesting enhanced ability to prioritize truly litigated patents in ranked outputs.

These empirical trends shown in Fig. 18 are consistent with theoretical expectations from probability calibration and statistical learning theory. When a model is trained on a

balanced dataset, it implicitly assumes a uniform class prior (*i.e.*, $P(y = 1) = 0.5$), which deviates significantly from the true prior observed in deployment scenarios. As a result, the model's estimates of the posterior probability $P(y = 1 \mid x)$ may become miscalibrated. According to Bayes' theorem, the true posterior is given by:

$$P(y = 1 \mid x) = \frac{P(x \mid y = 1) \cdot P(y = 1)}{P(x)}$$

where $P(y = 1)$ is the prior probability of litigation, $P(x \mid y = 1)$ is the likelihood of observing features $x$ given a litigated patent, and $P(x)$ is the marginal probability of observing $x$. When the model is trained using the correct class prior (*e.g.*, $P(y = 1) \approx 0.02$), the posterior probability estimation becomes more accurate, improving probability calibration and reducing the number of false positives.

From a risk minimization perspective, the goal is to minimize the expected loss over the true data distribution. The population risk is defined as:

$$\mathscr{R}(f) = \mathbb{E}_{(x,y) \sim P(x,y)}[\mathscr{L}(f(x), y)]$$

where $f(x)$ is the model's prediction for input $x$, $y \in \{0, 1\}$ is the true class label, and $\mathscr{L}(f(x), y)$ is a loss function that penalizes incorrect predictions. When training on artificially balanced datasets, the empirical risk diverges from the population risk, resulting in a biased optimization objective. As the training distribution aligns more closely with the true class distribution, the empirical risk becomes a better approximation of the true risk, yielding more generalizable models.

Improvements in threshold-independent metrics such as AP and Precision@200 further support the model's improved ranking capability. These metrics are particularly relevant in practical scenarios such as triaging or screening large patent portfolios, where ranking high-risk cases is more actionable than producing binary predictions.

Taken together, these findings underscore the value of aligning training data distributions with real-world class priors. Empirical results and theoretical insights demonstrate that aligning the training distribution with the true class priors leads to more accurate, calibrated, and useful predictions for downstream litigation risk assessment. These findings motivate future work on cost-sensitive training and dynamic class-weighting to further improve model robustness under deployment conditions.

## CONCLUSION

The scarcity of patent scope indicators based on the semantics of patent claim text is addressed in the first part of this study through the development of a new claim scope indicator, hyponym tree score (HTS). HTS utilizes the number of hyponyms of the words in a patent claim sentence, the sentence structure, and the inter-dependency among the patent claims in its calculation, as depicted in Algorithm 1. The final candidate for the HTS is selected from six computational options following a series of experiments that evaluate the performance improvements resulting from the inclusion of each HTS candidate in the litigation prediction task, as well as the results of the extremes study, information gain, and

feature correlation. A higher HTS value indicates a broader claim scope, hinting at higher legal coverage, increased value, increased litigation probability, and decreased patent grant probability. The second part of this study focuses on the development of a high-performing litigation prediction model suitable for predicting the litigation risk of patent drafts. A multifeature fusion approach is adopted to design the proposed MAPRA model, ensuring claim text understanding through a pre-trained model and augmenting it with additional numerical features. In the MAPRA model design, a BERT model is used for capturing claim text semantics, while numerical features such as HTS and other early-stage indicators are concatenated with the BERT output to improve litigation prediction. The MAPRA model achieves an AUC score of 0.878, surpassing the closest existing litigation prediction model, which is designed for granted patents and reports an AUC of 0.822. Given that MAPRA relies solely on pre-grant features available at the draft stage, this superior performance highlights its effectiveness and suitability for predicting litigation risk in both patent drafts and granted patents. It is suggested that patent authors can strategically manage the scope of claims during the drafting stage by leveraging HTS and MAPRA. The utilization of HTS and MAPRA enables authors to define claim boundaries precisely, thereby assisting patent examiners in efficiently identifying overly broad applications. For patent portfolio managers, HTS and MAPRA provide valuable insights for accurately assessing portfolio value and potential litigation risks. Furthermore, this model supports insurance companies in evaluating the litigation risks associated with newly granted patents, contributing to a more efficient, transparent, and well-regulated patent ecosystem.

In this study, hyponym counting of words relies on WordNet, which does not cover most scientific or domain-specific terms. Developing a context-aware hyponym *corpus* that includes technical and domain-specific terminology remains an important direction for future research. Recent work on LLM-based hyponym generation (*Yun et al., 2023*) provides promising insights for advancing such *corpus* development.

Patents are granted across a wide range of domains, each falling under different sections. The linguistic diversity inherent in documenting innovations from these varied fields necessitates a claim scope evaluation that is specific to each patent section. Such section-specific analysis could improve the quality of HTS and enhance litigation prediction performance for particular domains. This study refrains from recommending or defining specific HTS value ranges that may indicate claim scope boundaries. Establishing such recommendations would require section-specific analyses based on significantly larger datasets that reflect the true class distribution. Increasing dataset size with realistic class distribution and conducting patent section-specific evaluations represent key areas for future work. Additionally, this study does not account for temporal changes in patent litigation risk, which is another area for improvement (*Kim et al., 2021*). As there are currently no definitive or final litigation prediction models, there remains substantial scope for developing improved models, particularly those that incorporate enhanced hyponym corpora.

## ACKNOWLEDGEMENTS

The authors thank Mrs. Hélène Dernis, Analyst/Statistician, and the STI Microdata Lab at the Directorate for Science, Technology, and Industry, Organisation for Economic Co-operation and Development (OECD), France, for their assistance in ensuring seamless access to the PQI dataset. Additionally, the authors acknowledge the use of Grammarly for grammar and style checks and ChatGPT, developed by OpenAI, for language refinement and improving sentence structure during the preparation of this manuscript. These tools were used solely to enhance the readability and clarity of the text and did not influence the content, analysis, or conclusions of this research.

### Funding

The authors received no funding for this work.

### Competing Interests

The authors declare that they have no competing interests.

### Author Contributions

- Chitrakala Sakthivel conceived and designed the experiments, performed the experiments, analyzed the data, prepared figures and/or tables, authored or reviewed drafts of the article, and approved the final draft.
- Jinesh Jose conceived and designed the experiments, performed the experiments, analyzed the data, performed the computation work, prepared figures and/or tables, authored or reviewed drafts of the article, and approved the final draft.

### Data Availability

U.S. Patent and Trademark Office—Patent Litigation Docket Reports are available at: https://www.uspto.gov/ip-policy/economic-research/research-datasets/patent-litigation-docket-reports-data.

The Organisation for Economic Co-operation and Development (OECD)—Patent Quality Indicators Database (January 2024) is available at https://www.oecd.org/en/data/datasets/intellectual-property-statistics.html.

The U.S. Patent and Trademark Office—Patents View Dataset are available at: https://patentsview.org/download/data-download-tables.

The sampled dataset used for the experiments is available at Zenodo: Jose, J. (2025). Data used for the research on Patent Litigation Risk Prediction [Data set]. Zenodo. https://doi.org/10.5281/zenodo.14636279.

The source code and a user guide with step-by-step instructions for reproducing the results are available in File S1, and the computed claim scope values are available in File S2.

## Supplemental Information

Supplemental information for this article can be found online at http://dx.doi.org/10.7717/peerj-cs.3069#supplemental-information.

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
