# Peer review of "Multifeature fusion for claim scope-aware litigation risk prediction for patent drafts"

_PeerJ Computer Science, doi:10.7717/peerj-cs.3069_

## Round 0.1 · original submission · Major Revisions

Please respond to both reviewers.

·

Basic reporting

The manuscript is written in clear English overall, but I think there are some issues with the dataset description and the literature review.

1: The section on the dataset only briefly mentions the sources (e.g., USPTO, OECD, and litigation data) without giving details such as sample size, time range, preprocessing steps, or possible biases.
Please add more details about the dataset with key numbers and explanations so that readers can fully understand the data and its limitations.

2: Although many studies are cited, the literature review is not well organized and lacks a clear flow. This makes it hard to see what new ideas your work brings.
Please reorganize the literature review by grouping similar studies together and clearly pointing out the gaps in current research that your work addresses.

3: I noticed that the manuscript title uses the term "multimodal" to describe the mix of text and numerical features. Strictly speaking, this should be called "multifeature fusion."

Please clarify the term to match the standard usage in the field.

Experimental design

The HTS indicator uses WordNet hyponym counts and dependency tree structures to quantify the scope of claim texts. However, it mainly focuses on the semantic relationships within a single claim, which may overlook the inherent dependencies or associations among different claims.

1: How does the construction of HTS effectively capture the semantic characteristics of technical and legal terms in the patent texts?

2: Why was the dependency or relationship between different claims not modeled in the HTS? Could you explain the reasoning behind this decision?

3: The manuscript addresses the extreme imbalance by adopting a 1:1 sampling strategy, pairing each litigated patent with a non-litigated one. This may introduce a sampling bias that does not reflect the true distribution.

4: Patent litigation typically requires a technical comparison with reference documents to determine whether infringement issues exist between different patent claims. In the experiments, the authors only perform a binary classification on the patent text to decide whether litigation is likely. I believe this task setup needs to be more convincing. If a patent is deemed to have litigation risk by MAPRA, which patents serve as the reference documents? For legal applications, I think the results require stronger interpretability.

Validity of the findings

The MAPRA model's prediction improvement is marginal, and the model overall functions as a black box, lacking explanations for the contributions of individual features.

Please discuss the practical significance of this slight improvement and add feature importance analysis to enhance the model's credibility in real-world applications.

Reviewer 2 ·

Basic reporting

-

Experimental design

-

Validity of the findings

-

Additional comments

The text reads smoothly, and there's nothing to comment on.

Major Concern Regarding Sampling Methodology.

While the study makes valuable contributions through the novel HTS metric and MAPRA model, the 1:1 balanced sampling approach raises significant concerns about the validity of the litigation prediction results.

By artificially creating a balanced dataset (one litigated vs. one non-litigated patent) when the natural litigation rate is <2% (Chien, 2011; Wongchaisuwat et al., 2017), the evaluation fundamentally distorts the real-world prediction scenario. This approach essentially converts an extreme class imbalance problem (98:2 ratio) into an artificially simplified binary classification task, which may dramatically overestimate model performance.

The critical issue is that while balanced sampling can be justified for training (to prevent model bias), applying this to the test set fails to evaluate how the model would perform on the naturally imbalanced distribution it would encounter in practice. In real patent portfolios, the model would need to identify the rare litigated cases (needles) in a haystack of non-litigated patents. The current evaluation doesn't reflect this operational reality.

So the authors should report performance metrics (precision, recall, F1) on an imbalanced test set reflecting the natural 2% litigation rate.

---

## Round 0.2 · accepted · Accept

Thanks to the authors for their efforts to improve the work. Your revision satisfied the reviewer successfully. I believe it is ready for acceptance. Congratulations!

·

Basic reporting

The authors responded well to the reviewers comments and I consider this paper acceptable for publication.

Experimental design

The authors responded well to the reviewers comments and I consider this paper acceptable for publication.

Validity of the findings

The authors responded well to the reviewers comments and I consider this paper acceptable for publication.